# RAB23 coordinates early osteogenesis by repressing FGF10-pERK1/2 and GLI1

**Md Rakibul Hasan[1], Maarit Takatalo[1], Hongqiang Ma[1], Ritva Rice[1], Tuija Mustonen[1], David PC Rice[1,2]\***

[1]Craniofacial Development and Malformations research group, Orthodontics, Oral and Maxillofacial Diseases, University of Helsinki, Helsinki, Finland; [2]Oral and Maxillofacial Diseases, Helsinki University Hospital, Helsinki, Finland

**Abstract** Mutations in the gene encoding *Ras-associated binding protein 23* (*RAB23*) cause Carpenter Syndrome, which is characterized by multiple developmental abnormalities including polysyndactyly and defects in skull morphogenesis. To understand how RAB23 regulates skull development, we generated *Rab23*-deficient mice that survive to an age where skeletal development can be studied. Along with polysyndactyly, these mice exhibit premature fusion of multiple sutures resultant from aberrant osteoprogenitor proliferation and elevated osteogenesis in the suture. FGF10-driven FGFR1 signaling is elevated in *Rab23*[-/-] sutures with a consequent imbalance in MAPK, Hedgehog signaling and RUNX2 expression. Inhibition of elevated pERK1/2 signaling results in the normalization of osteoprogenitor proliferation with a concomitant reduction of osteogenic gene expression, and prevention of craniosynostosis. Our results suggest a novel role for RAB23 as an upstream negative regulator of both FGFR and canonical Hh-GLI1 signaling, and additionally in the non-canonical regulation of GLI1 through pERK1/2.

## Introduction

Ras-associated binding protein 23 (RAB23) belongs to the RAB family of small GTPases, which function during several steps in cell vesicle trafficking (*Evans et al., 2003*). RAB23 has 35–40% sequence homology with other RABs, it localizes to the plasma membrane and is proposed to regulate cargo internalization and relocation in the cell (*Evans et al., 2003*; *Leaf and Von Zastrow, 2015*; *Lim and Tang, 2015*; *Olkkonen et al., 1994*). RAB23 is unusual amongst RABs as it is involved in embryogenesis, for instance in heart and limb patterning, neural tube closure and skeletal development (*Eggenschwiler and Anderson, 2000*; *Eggenschwiler et al., 2001*; *Fuller et al., 2014*; *Günther et al., 1994*; *Hor and Goh, 2018*). RAB23 negatively regulates Hh signaling during mouse neural tube development (*Eggenschwiler and Anderson, 2000*), and it controls nodal signaling independently of Hh signaling during vertebrate left-right patterning (*Fuller et al., 2014*). RAB23 is involved in trafficking D1-type dopaminergic receptors (*Leaf and Von Zastrow, 2015*) and kinesin-2 motor protein Kif-17 (*Lim and Tang, 2015*) to the primary cilium. However, direct ciliary function of RAB23 is unclear as cilium length and function are reported normal in *Rab23*[-/-] mice (*Evans et al., 2003*; *Fuller et al., 2014*).

Multiple mutations in *RAB23* have been reported in patients with the autosomal recessive Carpenter syndrome; the majority of mutations result in nonsense-mediated decay while others include missense and in-frame deletions (*Jenkins et al., 2011*). Carpenter syndrome (CS, MIM #201000) or Acrocephalopolysyndactyly type II is a multi-organ developmental disorder with polysyndactyly, congenital heart defects, mental retardation, obesity and craniosynostosis as central features (*Carpenter, 1909*; *Jenkins et al., 2007*).

Craniosynostosis is the premature fusion of one or more craniofacial sutures that results in major disruption of face and skull growth. Mesenchymal cells in the center of the suture must be kept in an

**\*For correspondence:**
david.rice@helsinki.fi

**Competing interests:** The authors declare that no competing interests exist.

**eLife digest** In many animals, the skull is made of several separate bones that are loosely joined during childhood and only fuse into one piece when the animal stops growing. A genetic disease called Carpenter syndrome causes the bones of the skull to fuse early in life, stopping it from growing correctly. Carpenter syndrome is often caused by changes to the gene responsible for making a protein called RAB23.

RAB23 helps move other molecules and cell components between different parts of the cell, and is therefore involved in a number of cellular processes. Previous studies suggest that RAB23 has a role in many parts of the body during development. Yet, it is unclear which cells in the skull depend on RAB23 activity and how this protein is controlled.

To answer this question, Hasan et al. grew pieces of developing skull bones that had been taken from mice lacking the RAB23 protein in the laboratory. Examining these samples revealed that RAB23 is active in cells called osteoblasts that add new bone to the edge of each piece of the skull as it grows. Hasan et al. also found that RAB23 regulates two cellular signaling pathways – called the hedgehog pathway and the fibroblast growth factor pathway – that interact with one another and co-ordinate skull development.

These findings show how RAB23 controls the growth and fusion of skull bones in developing animals. This could improve our understanding of the role RAB23 plays in other processes during development. It also sheds light on the mechanisms of Carpenter syndrome which may inform new approaches for treating patients.

undifferentiated state to maintain suture patency, while progenitor cells at the osteogenic fronts proliferate and differentiate to facilitate bone growth. Suture biogenesis is dependent on the correct patterning of the skeletal elements, as well as the regulation of the mesenchymal stem cell niche, osteogenic condensation formation, osteoprogenitor proliferation and differentiation (*Rice and Rice, 2008*; *Twigg and Wilkie, 2015*). These developmental processes are regulated to permit coordinated craniofacial growth, without the fusion of the neighboring bones and consequent cessation of growth.

Mutations in *FGFR*s are a major cause of syndromic craniosynostosis, with many of the mutations conferring a gain of function (*Hajihosseini, 2008*; *Wilkie et al., 2017*). FGFRs isoforms show a distinct FGF ligand binding affinity and biological functions (*Ornitz and Itoh, 2015*). However, studies have shown that in an Apert syndrome (*FGFR2* mutation) mouse model isoform FGFR2c loses ligand specificity, and is able to bind with cognate FGF10 (*Ibrahimi et al., 2001*; *Ibrahimi et al., 2004*; *Johnson and Wilkie, 2011*; *Yu et al., 2000*). Interestingly, genetic knockdown of *Fgf10* in this mouse model could rescue the premature fusion (*Hajihosseini et al., 2009*). FGF signaling pathway members have not been linked to RAB23-mediated trafficking. However, study suggests that RAB23 resides in the plasma membrane and proposed to be involved in endocytosis (*Evans et al., 2003*). In this context, RAB23 might have a direct role in growth factor receptor recycling and turnover, and therefore regulate the availability of the FGF receptors at the cell surface (*Langemeyer et al., 2018*; *Zerial and McBride, 2001*).

Similar to Carpenter syndrome, several craniosynostosis syndromes caused by mutations in *FGFRs* are characterized by patients exhibiting syndactyly and occasionally polysyndactyly (*Goos and Mathijssen, 2019*; *Mantilla-Capacho et al., 2005*). Also, mutations in several hedgehog (Hh) pathway members cause polydactyly (*Malik, 2014*; *Ullah et al., 2019*). Notably, Greig cephalopolysyndactyly syndrome (MIM # 175700) is caused by haploinsufficiency of the Hh signaling negative regulator, *GLI3* (*Vortkamp et al., 1991*). In addition to the polysyndactyly, some patients with Greig cephalopolysyndactyly syndrome exhibit craniosynostosis, and the mouse model for Greig cephalopolysyndactyly syndrome (*Gli3$^{Xt-J/Xt-J}$*) shows complete phenotypic penetrance for both craniosynostosis and polydactyly (*Hurst et al., 2011*; *Rice et al., 2010*). Also, a recurrent mosaic mutation in the hedgehog receptor *Smoothened* (*SMO*) causes craniosynostosis and polysyndactyly (*Twigg et al., 2016*). These overlapping skeletal phenotypes caused by mutations in *FGFRs*, Hh regulators and *RAB23* are suggestive of common etiological mechanisms. FGF and Hh signaling have well-defined roles during intramembranous osteogenesis. FGF signaling regulates many stages including

mesenchymal condensation formation, osteoprogenitor proliferation and differentiation and activation of the osteogenic transcription factor RUNX2 (*Debiais et al., 1998*; *Kim et al., 1998*; *Ornitz and Itoh, 2015*; *Yoon et al., 2014*). IHH positively regulates osteoprogenitor recruitment to the osteogenic front and GLI transcription factors regulate stem cell maintenance and osteoprogenitor proliferation (*Lenton et al., 2011*; *Rice et al., 2010*; *Veistinen et al., 2012*; *Zhao et al., 2015*). Interestingly, RAB23 regulates GLI1 in a Su(Fu)-dependent manner (*Chi et al., 2012*) and GLI1-positive cells have been identified in the suture as the main source of mesenchymal stem cells that plays crucial role in suture patency (*Zhao et al., 2015*).

The aim of this study was to determine the role of RAB23 during intramembranous bone development. Previously, it has not been possible to study skeletal development in RAB23 deficient mice due to their early lethality. Here, we generated RAB23 deficient (*Rab23*[-/-]) mice that survived until embryonic day (E) 18.5, and this allowed us to investigate how RAB23 regulates bone formation. Similar to Carpenter syndrome patients, RAB23 deficient mice exhibited polysyndactyly and multiple craniosynostoses. We show that disruption of *Rab23* leads to an upregulation of *Pitx2*, *Fgf10* and *Fgfr1b* expression, decreased p38 and enhanced pERK1/2-RUNX2 signaling along with elevated osteoprogenitor proliferation. In addition, *Hh* signaling was amplified with increased expression of GLI1. During in vitro culture, inhibition of elevated pERK1/2 normalized osteoprogenitor proliferation, corrected the aberrant GLI1 and RUNX2 expressions, and rescued the lambdoid suture fusion. Our results suggest a novel role for RAB23 as an upstream regulator of both FGF10-pERK1/2 and Hh-GLI1, and the additional regulation of GLI1 by pERK1/2, to coordinate the initiation of osteogenesis.

## Results

### *Rab23*[-/-] mice exhibit craniosynostosis in multiple sutures

*Rab23 open brain 2 (opb2)* homozygous mutant mice did not produce RAB23 protein in primary cells, analyzed by western blot (*Figure 1—figure supplement 1*) and will therefore be referred to as *Rab23*[-/-] mice. *Rab23*[-/-] C3 Heb/FeJ mice, generated through an ENU screen, survive until E12.5 rendering full analysis of organogenesis impossible (*Eggenschwiler et al., 2001*; *Kasarskis et al., 1998*). To obviate the gestation lethality before skeletal development, we backcrossed *Rab23*[-/-] C3 Heb/FeJ mice onto the C57Bl/6 strain. After six generations, the survival of the *Rab23*[-/-] homozygote embryos were prolonged to E18.5, and this allowed us to study skeletal development of these mice.

*Rab23*[-/-] mice at embryonic day E18.5 exhibited craniosynostosis in the coronal, parietal-temporal, fronto-nasal and in the lambdoid sutures when compared to their wild type (Wt) littermates (*Figure 1A–E*). The prevalence of premature suture fusion in *Rab23*[-/-] mice represented as percentage and only parieto-temporal sutures showed bi-lateral suture fusion (100%) in all the samples (*Figure 1F*). Along with craniosynostosis, *Rab23*[-/-] mice showed skeletal patterning defects in the forelimbs and hindlimbs. *Rab23*[-/-] mice showed pre-axial polydactyly (seven digits) of the forelimb and preaxial polysyndactyly (seven digits) of the hindlimbs (*Figure 1G*). Polydactyly and craniosynostosis were observed (100%) in all *Rab23*[-/-] samples examined.

At E18.5, 5% of *Rab23*[-/-] lambdoid sutures exhibited unilateral suture fusion (*Figure 1F*). Compared to Wt samples (*Figure 1I*), unfused *Rab23*[-/-] lambdoid sutures showed abnormal bony protrusions from parietal bones toward interparietal bone (*Figure 1J*) or ectopic bones in the lambdoid suture (*Figure 1K*). At this stage, *Rab23*[-/-] lambdoid sutures were narrower than Wt lambdoid sutures (*Figure 1L*). As 95% of *Rab23*[-/-] lambdoid sutures were consistently patent at E18.5 they were chosen as a model to study the role of RAB23 in osteogenesis. *Rab23*[-/-] mice die neonatally, therefore, analysis of *Rab23*[-/-] lambdoid suture beyond the E18.5 stage was assessed by in vitro calvarial explant culture (*Rice et al., 2003b*; *Figure 1M*). Wt and patent *Rab23*[-/-] lambdoid sutures harvested for culture at E18.5, *Rab23*[-/-] lambdoid sutures fused predictably after 3 days of culture (*Figure 1N,O*).

### *Rab23* expression in calvaria and sutures

*Rab23* mRNA expression was analyzed in whole calvaria and tissue sections by in situ hybridization and protein expression by immunohistochemical staining and western blotting. *Rab23* was expressed in suture mesenchyme with the strongest signal in interfrontal, sagittal and coronal

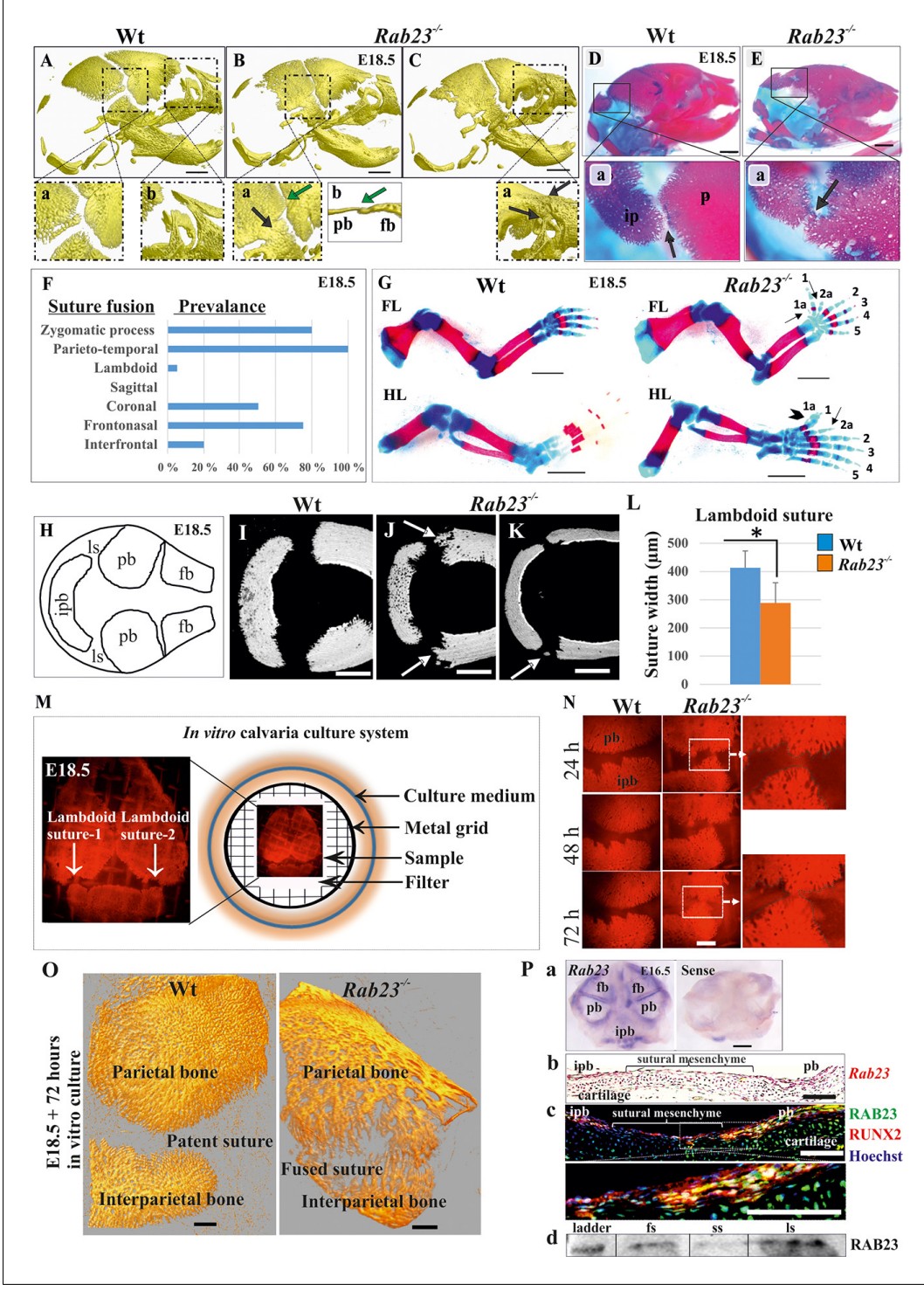

**Figure 1.** Deficiency of RAB23 causes premature fusion of multiple sutures and polysyndactyly. (A–E) Analysis of Wt and *Rab23*⁻/⁻ skulls by μ-CT (A–C) and alizarin red, alcian blue staining (D–E) at E18.5. *Rab23*⁻/⁻ skulls show fusion in the parieto-temporal suture (B-a, black arrow), coronal suture (B-a, green arrow and B-b, μ-CT slice, green arrow), frontonasal suture (C-a, arrow) and lambdoid suture (E-a, arrow). Wt sutures were open at this embryonic stage (A–a, A–b) (n = 10 for each age and genotype). fb: frontal bone, pb: parietal bone. Scale bar: 1 mm. (F) The prevalence of suture fusion in *Rab23*⁻/⁻ mouse at E18.5 shown in percentage. Only parieto-temporal suture showed bi-lateral suture fusion in all the samples. (n = 10 for sagittal and interfrontal suture, n = 40 for lambdoid suture and n = 20 for other sutures). (G) Skeletal Analysis of the limbs in *Rab23*⁻/⁻ mouse show pre-axial

*Figure 1 continued on next page*

*Figure 1 continued*

polydactyly of the fore limb (FL, 1a, 2a) and pre axial polysyndactyly of the hind limb (HL, 1a, 2a) at E18.5 (n = 10 for each age and genotype). Scale bar: 1.5 mm. (**H–K**) E18.5 mouse calvaria indicating fb: frontal bone, pb: parietal bone, ipb: interparietal bone and ls: lambdoid suture (**H**). Analysis of Wt and *Rab23<sup>-/-</sup>* calvaria by μ-CT at E18.5 shows *Rab23<sup>-/-</sup>* lambdoid sutures form bony protrusions from parietal bones project towards the interparietal bones (J, arrow), or ectopic bony islands in the mid-sutural mesenchyme (K, arrow) (n = 6 for each age and genotype). Scale bar: 500 μm. (**L**) Measurements of the lambdoid suture shows *Rab23<sup>-/-</sup>* lambdoid sutures are narrower as compare to the Wt samples at E18.5 (n = 6 for each age and genotype). Data represented as mean ± SD, paired Student's *t*-test was used. Statistical significance was defined as a p-*value* < 0.05 (*). (**M**) In vitro calvaria culture system. This system was used to culture Wt and *Rab23<sup>-/-</sup>* calvaria containing patent lambdoid sutures at E17.5 and E18.5. (**N, O**) Represents E18.5 Wt and patent lambdoid suture containing *Rab23<sup>-/-</sup>* calvaria culture in vitro for 3 days in presence of alizarin red. *Rab23<sup>-/-</sup>* lambdoid suture shows fusion at day 3 (N, alizarin red bone staining, O, μ-CT images), whereas Wt controls remain open (n = 10 for each genotype). Scale bar: 500 μm. (**P**) *Rab23* expression in the whole calvarial tissue at E16.5 is shown by digoxigenin labeled whole mount in situ hybridization (**P–a**) and shown at E15.5 sutural tissue sections by RNAscope (**P–b**). Co-expression of RAB23 (green) and osteoblast marker RUNX2 (red) in the calvarial sutural section at E17.5 is shown by immunohistochemical staining (**P–c**), nuclear staining (blue). RAB23 protein expression in the sutures at E15.5 is shown by western blotting (**P–d**). fb: frontal bone, pb: parietal bone, ipb: interparietal bone, fs: frontal suture, ss: sagittal suture, ls: lambdoid suture. Scale bar: 500 μm (a), 100 μm (b, c).

The online version of this article includes the following figure supplement(s) for figure 1:

**Figure supplement 1.** RAB23 protein expression in WT and *Rab23<sup>-/-</sup>* mouse calvaria derived primary cells.

sutures and parietal bone side of the lambdoid suture (*Figure 1P–a*). Analysis of tissue sections revealed *Rab23* expression in the osteogenic fronts of the calvarial sutures and throughout the sutural mesenchyme (*Figure 1P–b*). RAB23 was expressed in the lambdoid sutural osteogenic fronts and in the underlying cartilage. Co-expression with RUNX2 confirmed the cells in the osteogenic front to be osteoprogenitor or osteoblast (*Figure 1P–c*). The *Rab23* expression pattern was consistent with the phenotype of synostosed calvarial sutures observed in the *Rab23<sup>-/-</sup>* mice (*Figure 1B,C and E*). Western blotting of proteins extracted from E15.5 Wt calvarial frontal, sagittal and lambdoid sutures along with their osteogenic fronts confirmed the presence of RAB23 protein in the suture (*Figure 1P–d*).

## *Rab23<sup>-/-</sup>* mice exhibit elevated levels of *Fgf10* and *Pitx2* in the calvaria and suture

To investigate differentially expressed genes between Wt and *Rab23<sup>-/-</sup>* calvaria and to find candidate genes responsible for the craniosynostosis in the RAB23-deficient mice, we performed a microarray-based gene expression analysis of whole Wt and *Rab23<sup>-/-</sup>* calvaria samples excluding the skin and underlying brain at E15.5 (*Figure 2A*). Our analysis revealed 223 significantly differentially expressed genes between the Wt and *Rab23<sup>-/-</sup>* calvaria (t-test, p<0.05) (*Figure 2B*, *Supplementary file 1*, MIAME-compliant data has been deposited in GEO database as source data, GEO accession GSE140884). Among these genes, 115 genes were upregulated (*Figure 2B*, red) and 108 genes were downregulated (*Figure 2B*, green). *Fgf10* was found to be overexpressed in the microarray in *Rab23<sup>-/-</sup>* calvaria (*Figure 2C*). *Fgf10* was selected as a candidate for further analysis as it has previously been shown to be expressed early in calvarial development and has been implicated in craniosynostosis pathogenesis (*Hajihosseini et al., 2009*; *Veistinen et al., 2009*). *Fgf10* expression was elevated in whole calvarial mesenchyme, lambdoid sutural mesenchymal tissue and in calvaria derived (CD) mesenchymal cells at E15.5 as assessed by RT-qPCR (*Figure 2D*). Our microarray analysis also showed that the transcription factor *Pitx2* was elevated in *Rab23<sup>-/-</sup>* calvaria (*Figure 2C*). *Pitx2* is a well-documented upstream regulator of *Fgf10* and may act in a reciprocal regulatory loop with FGF10 during early organogenesis (*Al Alam et al., 2012*). We found that *Pitx2* was overexpressed both in the *Rab23<sup>-/-</sup>* calvaria and in *Rab23<sup>-/-</sup>* lambdoid sutures (*Figure 2E*). *Pitx2* overexpression was

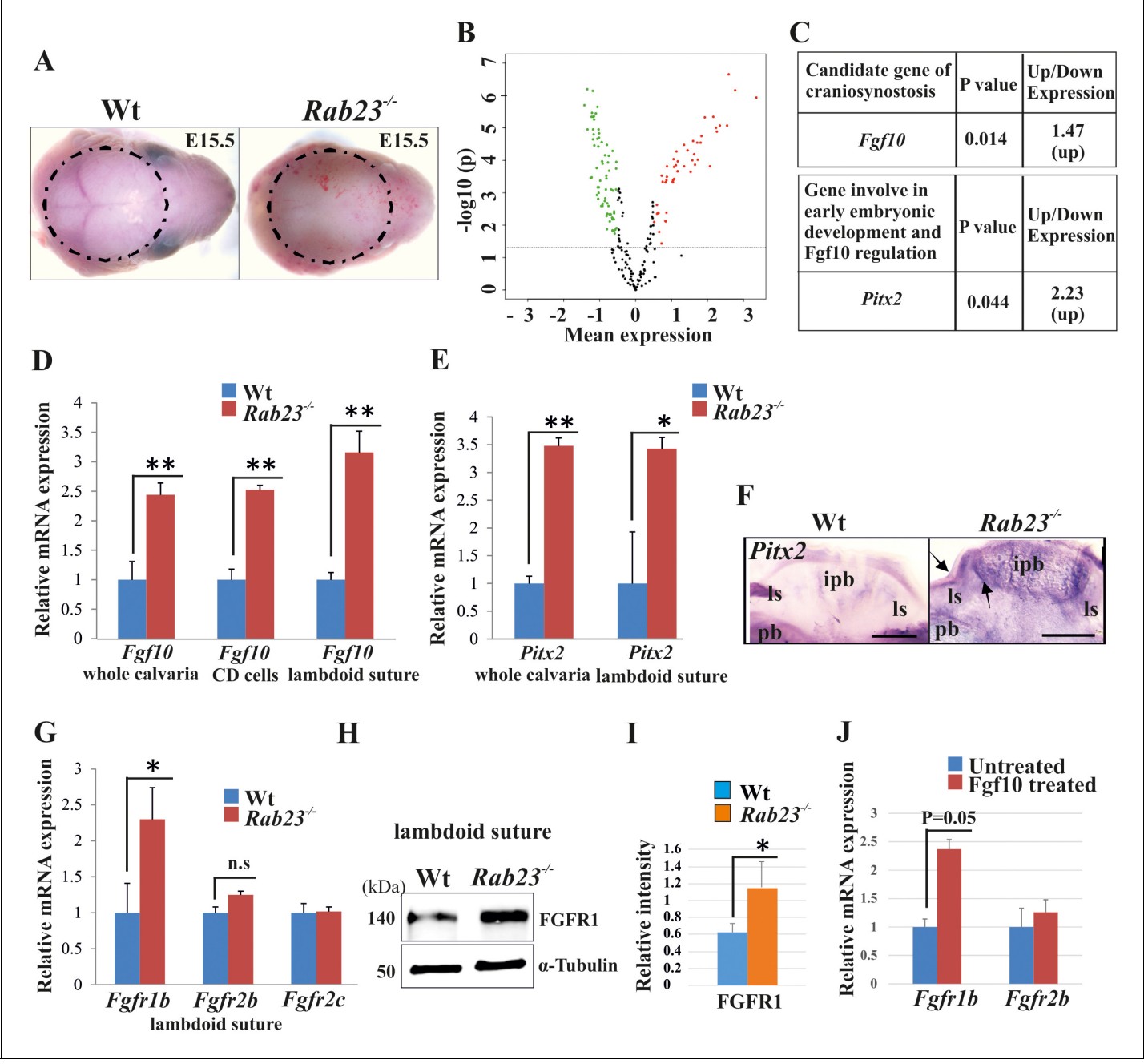

**Figure 2.** Gene expression analysis in Wt and *Rab23*−/− lambdoid suture. (**A**) Wt and *Rab23*−/− calvaria at E15.5 (dotted ring, excluding skin) processed for Illumina based microarray (n = 3 for each age and genotype). (**B**) Analysis of mRNA based microarray data by Chipster, a bioinformatics tool, reveals that 223 genes are differentially expressed (*t*-test, p<0.05, 115 genes are upregulated, red and 108 genes are downregulated, green) in *Rab23*−/− calvaria, represented by volcano plot (n = 3 for each age and genotype). Fold change of genes is calculated by arithmetic mean in linear scale and shown in the volcano plot. Fold change >1 (up-regulated gene), fold change < 1 (down-regulated gene). The gene list is provided in *Supplementary file 1*. (**C**) Represents individual searches of genes based on their contribution of suture fusion reveals *Fgf10* as a candidate gene and *Pitx2* as another developmentally important gene that can regulate *Fgf10* expression. Both genes are upregulated in *Rab23*−/− calvaria. (**D, E**) RT-qPCR analysis of *Fgf10* (**D**) and *Pitx2* (**E**) mRNA extracted from E15.5 Wt and *Rab23*−/− calvaria (excluding skin), cultured calvaria derived primary cells (CDC) and from lambdoid sutural tissue reveals that *Fgf10* and *Pitx2* are overexpressed in *Rab23*−/− samples (n = 3 calvaria and eight lambdoid sutures for each genotype). Gene expressions were normalized by *18S rRNA*. Data are represented as mean ± SD, paired Student's *t*-test was used and as relative gene expression is shown using ΔΔCτ values. Statistical significance was defined as a p-*value* <0.05 (*), p-*value* <0.005 (**). (**F**) *Pitx2* expression analysis by whole-mount ISH in Wt and *Rab23*−/− calvaria at E15.5. Arrows indicate *Pitx2* overexpression in the lambdoid suture. pb: parietal bone, ipb: interparietal bone, ls: lambdoid suture. Scale bar: 500 μm. (**G**) RT-qPCR expression analysis of *Fgfr1b*, *Fgfr2b* and *Fgfr2c* mRNA from E15.5 Wt and *Rab23*−/−

*Figure 2 continued on next page*

Figure 2 continued

lambdoid suture reveals *Fgfr1b* overexpression in *Rab23*[-/-] sample (n = 8 for each genotype). Gene expressions were normalized by *18S rRNA*. Data are represented as mean ± SD, paired Student's *t*-test was used and relative gene expression is shown using ΔΔCτ values. Statistical significance was defined as a *P-value* <0.05 (*). (H, I) Western blotting of proteins extracted from Wt and *Rab23*[-/-] lambdoid suture at E15.5 (H) and relative intensity measurement (I) reveals over expression of FGFR1 in the *Rab23*[-/-] lambdoid suture (n = 6 for each genotype). Data represented as mean ± SD, paired Student's *t*-test was used. Statistical significance was defined as a p-*value* <0.05 (*). (J) Exogenous FGF10 treatment for 3 hr on Wt calvaria derived (CD) cells and subsequent *Fgfr1b and Fgfr2b* mRNA analysis by q-PCR shows induction of *Fgfr1b* expression in FGF10 treated cells compare to untreated Wt CD cells (n = 3 for each genotype). Gene expressions were normalized by *18S rRNA*. Data are represented as mean ± SD, paired Student's *t*-test was used and relative gene expression is shown using ΔΔCτ values. Statistical significance was defined as a *P-value*.

The online version of this article includes the following figure supplement(s) for figure 2:

**Figure supplement 1.** FGFR1 expression in Wt and *Rab23*[-/-] lambdoid suture.

further observed in *Rab23*[-/-] calvaria by whole-mount in situ hybridization in and surrounding the interparietal bone (*Figure 2F*).

## FGF10-induced upregulation of *Fgfr1b* expression in *Rab23*[-/-] lambdoid suture

We assessed the expression of FGF10-specific receptors *Fgfr1b*, *Fgfr2b*, also *Fgfr2c* in Wt and *Rab23*[-/-] lambdoid sutural mesenchymal tissue. Our analysis showed elevated *Fgfr1b* expression and a non-significant upregulation of *Fgfr2b* in *Rab23*[-/-] samples (*Figure 2G*). FGFR1 protein levels were found to be elevated in *Rab23*[-/-] lambdoid sutures compared to Wt samples when analyzed by western blotting (*Figure 2H,I*) and by immunohistochemical staining (*Figure 2—figure supplement 1*). The expression of *Fgfr2c* was unchanged (*Figure 2G*). To test whether the upregulation of *Fgfr1b* and *Fgfr2b* expression was due to the high levels of FGF10 ligand present in the *Rab23*[-/-] lambdoid suture, we isolated E15.5 wild-type calvaria derived (CD) mesenchymal cells and treated them with exogenous FGF10 (250 ng/ml) for 3 hr. *Fgfr1b* and *Fgfr2b* expression levels were then assessed by RT-qPCR. We found a significant (p=0.05) upregulation of *Fgfr1b* expression and a slight upregulation of *Fgfr2b* expression in the cultured CD mesenchymal cells after exogenous FGF10 stimulation, indicating FGFR1b as a target of FGF10 in the calvarial mesenchyme (*Figure 2J*).

## Opposing MAPK-RUNX2 and Hh-GLI1 signaling in *Rab23*[-/-] lambdoid suture

FGF signaling has been shown to activate MAPK signaling pathway subtypes pERK and p38 in suture morphogenesis and osteoblast differentiation (*Kyono et al., 2012*; *Pfaff et al., 2016*; *Figure 3A*). In *Rab23*[-/-] lambdoid suture, we found decreased p38 and elevated pERK1/2 and RUNX2 expressions compare to Wt samples (*Figure 3B–G*). To test whether the upregulation of RUNX2 expression was due to the high levels of FGF10 present in the *Rab23*[-/-] lambdoid suture, we isolated E15.5 *Rab23*[-/-] calvaria derived (CD) mesenchymal cells and treated them with exogenous FGF10 (500 ng/ml) for 2 and 4 hr. *Runx2* expression levels were then assessed by RT-qPCR. We found a significant (p<0.05) upregulation of *Runx2* expression at both time points compare to untreated samples, indicating that RUNX2 as a target of FGF10 in the calvarial mesenchyme (*Figure 3H*). Since ERK1/2 signaling regulates Hh and GLI expression, and RAB23 is a known negative regulator of Hh signaling (*Eggenschwiler et al., 2001*), we further assayed *Hh, Gli1, Gli2, and Gli3* transcripts extracted from Wt and *Rab23*[-/-] lambdoid sutural mesenchyme at E15.5. We detected a dramatic increase in the expression of *Hh* (*Figure 3I*) and *Gli1* (*Figure 3J*), a sutural stem cell marker (*Zhao et al., 2015*). Although *Gli3* expression is not changed, a functional change in the ratio of GLI3FL to GLI3R may still possible (*Veistinen et al., 2017*). As TGFβ-superfamily members have also been shown to regulate suture patency (*Kim et al., 1998*; *Komatsu et al., 2013*; *Pan et al., 2017*), we assayed both pSMAD1/5/8 and pSMAD2/3 expressions but found no difference between Wt and mutant samples (*Figure 3—figure supplement 1*).

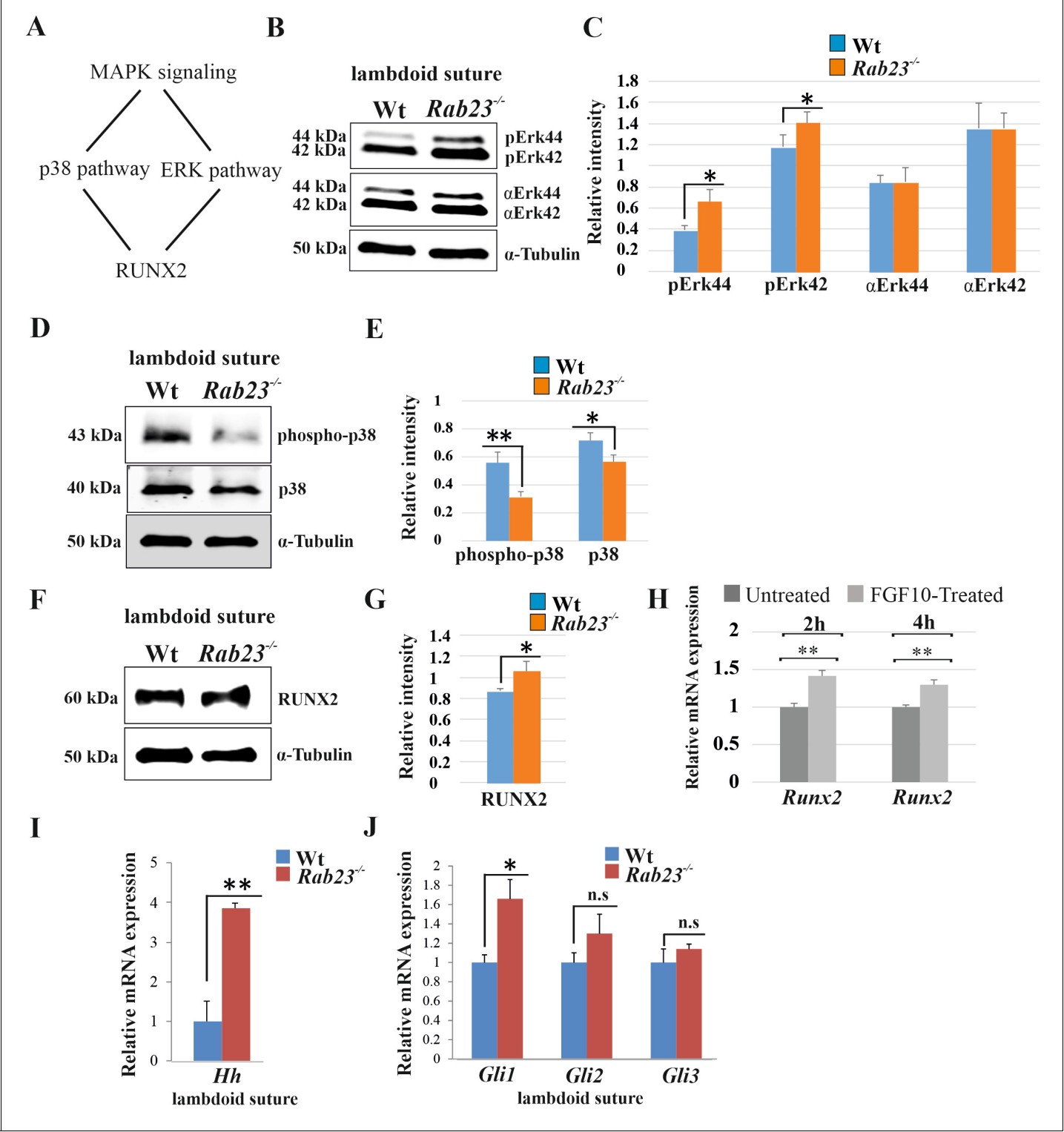

**Figure 3.** Analysis of MAPK signaling in Wt and *Rab23*[-/-] lambdoid suture. (A) Represents two downstream pathway subtypes p38 and ERK of MAPK signaling involve in RUNX2 activation and suture fusion. (B, C) Western blotting analysis of pERK44/42 and α-ERK44/42 protein levels extracted from Wt and *Rab23*[-/-] lambdoid suture at E15.5 (B) and relative intensity measurement (C) shows higher pERK44 and pERK42 levels in *Rab23*[-/-] lambdoid suture (n = 6 for each genotype). Data represented as mean ± SD, paired Student's *t*-test was used. Statistical significance was defined as a *P-value* <0.05 (*). (D, E) Western blotting analysis of phospho-p38 and p38 protein levels extracted from Wt and *Rab23*[-/-] lambdoid suture at E15.5 (D) and relative intensity measurement (E) shows lower phospho-p38 and p38 levels in *Rab23*[-/-] lambdoid suture (n = 6 for each genotype). Data represented as

*Figure 3 continued on next page*

*Figure 3 continued*

mean ± SD, paired Student's *t*-test was used. Statistical significance was defined as a p-*value* <0.05 (*), p-*value* <0.005 (**). (**F, G**) Western blotting analysis of RUNX2 protein level extracted from Wt and *Rab23*$^{-/-}$ lambdoid suture at E15.5 (**F**) and relative intensity measurement (**G**) shows higher RUNX2 level in *Rab23*$^{-/-}$ lambdoid suture (n = 6 for each genotype). Data represented as mean ± SD, paired Student's *t*-test was used. Statistical significance was defined as a p-*value* <0.05 (*). (**H**) *Runx2* expression analysis by RT-qPCR in exogenous FGF10 treated (500 ng/ml) and untreated *Rab23*$^{-/-}$ calvaria derived cells at 2 and 4 hr. (n = 3 for each genotype). Gene expressions were normalized by *18S rRNA*. Data are represented as mean ± SD, paired Student's *t*-test was used and relative gene expression is shown using ΔΔC$_T$ values. Statistical significance was defined as a *P-value* <0.05 (*), *P-value* <0.005 (**). (**I, J**) RT-qPCR expression analysis of hedgehog signaling components *Hh* (**H**), *Gli1*, *Gli2* and *Gli3* (**I**) in the Wt and *Rab23*$^{-/-}$ lambdoid suture reveals overexpression of *Hh* and *Gli1* in *Rab23*$^{-/-}$ lambdoid suture (n = 8 for each genotype). Gene expressions were normalized by *18S rRNA*. Data are represented as mean ± SD, paired Student's *t*-test was used and relative gene expression is shown using ΔΔC$_T$ values. Statistical significance was defined as a *P-value* <0.05 (*), *P-value* <0.005 (**). n.s: non-significant.

The online version of this article includes the following figure supplement(s) for figure 3:

**Figure supplement 1.** pSMAD1/5/8 and pSMAD2/3 expression in Wt and *Rab23*$^{-/-}$ lambdoid suture.

## *Rab23*$^{-/-}$ lambdoid suture and MEF cells show increased osteoprogenitor cell proliferation

As FGF10-pERK1/2 signaling and RUNX2 are known to regulate suture osteoprogenitor proliferation (*Kim et al., 1998*; *Qin et al., 2019*), we assessed co-expression of EdU and RUNX2 in the WT and *Rab23*$^{-/-}$ lambdoid suture. EdU and RUNX2 co-localized in the osteogenic fronts indicating that the proliferating cells are osteoprogenitors (*Figure 4A,B*). We analyzed the proliferation in Wt and *Rab23*$^{-/-}$ lambdoid sutures (*Figure 4C–H*). EdU pulsed *Rab23*$^{-/-}$ samples showed more proliferating cells in the osteogenic fronts compared to Wt (*Figure 4C–H*). In addition, increased level of cell proliferation was observed in *Rab23*$^{-/-}$ derived mouse embryonic fibroblast (MEF) cells (*Figure 4I*). *Rab23*$^{-/-}$ calvaria derived cells also had an elevated number of total cells at different passages (P$^1$ to P$^3$) when compared to Wt calvaria derived cells (*Figure 4J*). Increased proliferation in the sutures can result in a larger pool of osteoprogenitor cells and enhanced osteogenesis (*Lana-Elola et al., 2007*; *Rice et al., 2003b*). Taken together, this data shows that RAB23 is needed to repress proliferation in undifferentiated mesenchymal cells.

## RAB23 regulates suture patency through a MEK-driven mechanism

Our data suggest that *Rab23*$^{-/-}$ lambdoid suture undergoes fusion through aberrant FGF signaling, specifically by activating pERK1/2 signaling, RUNX2 and enhanced cell proliferation of osteoprogenitor cells. To test this, we aimed to normalize the *Rab23*$^{-/-}$ lambdoid suture by inhibiting the activation of MEK1/2, a downstream component of the MAPK-ERK1/2 signaling pathway by using MEK1/2 inhibitor U0126. We cultured E18.5 Wt and *Rab23*$^{-/-}$ whole calvaria in osteogenic growth medium up to 72 hr. To confirm that lambdoid sutures were patent at time point zero, the vital bone dye calcein green was used to detect the bone edges. Calcein green was administered to pregnant females 24 hr before samples were taken. Half of the *Rab23*$^{-/-}$ calvaria were treated with U0126 and half of the *Rab23*$^{-/-}$ calvaria samples were untreated (*Figure 5A,B*). None of the untreated *Rab23*$^{-/-}$ (n = 0/12) and 84.34% of U0126 treated *Rab23*$^{-/-}$ (n = 11/12) lambdoid sutures were rescued (*Figure 5A,B*). To confirm whether bones were fused or not, Wt, U0126 treated and untreated *Rab23*$^{-/-}$ samples were further analyzed in tissue sections and by μ-CT (*Figure 5A–C*, *Figure 5—figure supplement 1* and *Figure 5—Videos 1–3*). None of the Wt lambdoid sutures were fused (n = 0/12). The same experiment (n = 4 in each genotype) was carried out with E17.5 samples and cultured for 4 days. Untreated *Rab23*$^{-/-}$ lambdoid sutures underwent suture fusion by day 4, while U0126-treated *Rab23*$^{-/-}$ lambdoid sutures remained patent during the culture period (*Figure 5—figure supplement 2*). The effect of U0126 on sutural cells was also analyzed by cell death assay and no effect was observed in U0126-treated lambdoid sutural cells (*Figure 5—figure supplement 3*). U0126 treatment not only prevented suture fusion but it also normalized the level of osteoprogenitor cell proliferation (*Figure 5D–G*). EdU incorporation revealed that U0126 treated *Rab23*$^{-/-}$ lambdoid sutures (n = 8)

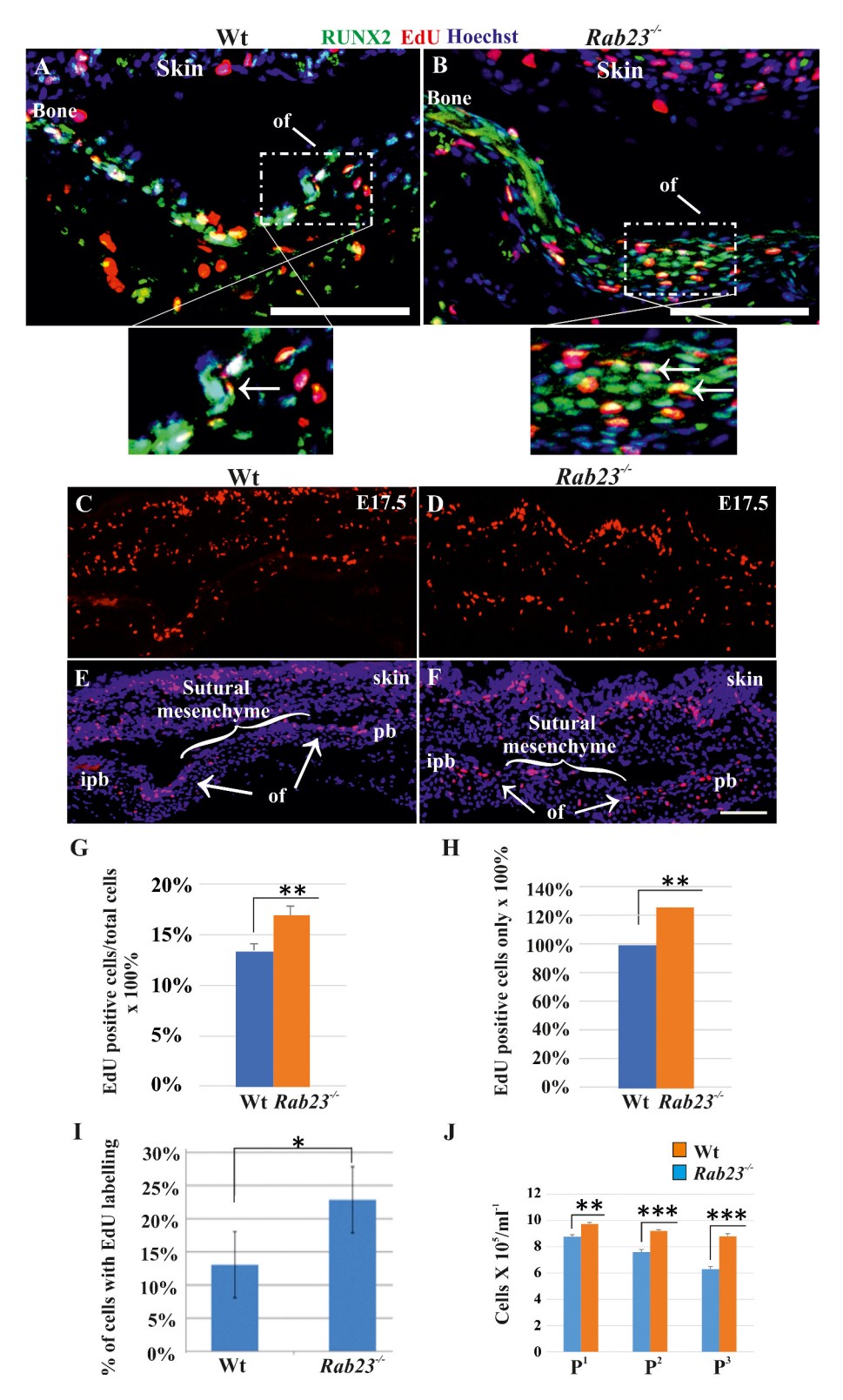

**Figure 4.** Cell proliferation analysis in Wt and *Rab23*[-/-] lambdoid sutural cells and MEF cells. (A–H) EdU pulsed assay in the Wt and *Rab23*[-/-] lambdoid sutures at E17.5 shows proliferating cells in red color (A–F). Proliferating cells show co-localization with osteoprogenitor and osteoblast marker RUNX2 (green) in the osteogenic front (inset, white arrow) in both Wt and *Rab23*[-/-] lambdoid suture (A, B). Analysis of EdU pulsed cells (C–F) together with nuclear staining (E, F) in the Wt and *Rab23*[-/-] lambdoid suture and subsequent quantification revealed that *Rab23*[-/-] sutures show higher cell

*Figure 4 continued on next page*

*Figure 4 continued*

proliferation as percentage of EdU-positive cells compare to total cells (**G**) and percentage of EdU-positive cells only (**H**) in those sutures are higher compare to Wt samples (n = 4 for each genotype). Data represented as mean ± SD, paired Student's *t*-test was used. Statistical significance was defined as a *P*-value <0.005 (**). pb: parietal bone, ipb: interparietal bone, ls: lambdoid suture, of: osteogenic front. Scale bar: 100 μm. (**I**) EdU incorporation in the cultured Wt and *Rab23⁻/⁻* MEF cells isolated from E13.5 embryos show 8–15% more cell proliferation (DNA duplication) in *Rab23⁻/⁻* samples compare to corresponding Wt samples (n = 3 for each genotype). Data represented as mean ± SD, paired Student's *t*-test was used. Statistical significance was defined as a p-*value* <0.05 (*). (**J**) Represents passaging (P¹ to P³ ) of Wt and *Rab23⁻/⁻* calvaria derived primary cells in the culture show the total number of *Rab23⁻/⁻* cells (ml⁻¹) in each passage increases more rapidly than Wt cells, while the cell viability (determined by trypan blue) and cell size were similar in all cell lines. (n = 3 for each genotype). Data represented as mean ± SD, paired Student's *t*-test was used. Statistical significance was defined as a p-value <0.005 (**), p-*value* <0.001 (***).

exhibited reduced proliferation when compared to *Rab23⁻/⁻* controls (n = 8) (***Figure 5D–G***) and normalized it to Wt levels (n = 8) after 24 hr of culture (***Figure 5D,E***). The proliferation in all genotypes decreased over time (***Figure 5D–G***). This indicates that RAB23 regulates suture patency through a MAPK-MEK-ERK-driven mechanism. pERK1/2 regulates RUNX2 and GLI1 expression in calvaria derived mesenchymal cells.

We analyzed the effect of the MEK1/2 inhibitor U0126, cyclopamine and the combined action of cyclopamine and U0126 on pERK1/2, RUNX2 and GLI1 expression levels in *Rab23⁻/⁻* CD cells at 6, 24 and 48 hr of culture (***Figure 6***). After 6 hr of culture, the relative pERK1/2 levels were significantly reduced in U0126-treated *Rab23⁻/⁻* cells compare to untreated *Rab23⁻/⁻* cells. The downregulation of pERK1/2 continued in U0126-treated *Rab23⁻/⁻* cells at 24 and 48 hr of culture compare to the untreated *Rab23⁻/⁻* cells (***Figure 6A,B***). Whereas, RUNX2 expression significantly reduced only after 48 hr of U0126-treated *Rab23⁻/⁻* cells (***Figure 6A,C***). U0126-treated *Rab23⁻/⁻* cells showed gradual reduction of GLI1 expressions at 6 and 24 hr; however, it showed drastic reduction at 48 hr compared to untreated *Rab23⁻/⁻* cells (***Figure 6A,D***). Further immunohistochemical expression of GLI1 was also found reduced in the U0126 treated *Rab23⁻/⁻* lambdoid sutures at 24 and 48 hr compared to untreated *Rab23⁻/⁻* lambdoid sutures (***Figure 6—figure supplement 1***). The effect of cyclopamine and combined action of cyclopamine and U0126 on these cells showed that cyclopamine has no effect on pERK1/2 expression and can reduce RUNX2 expression only after prolonged exposure (48 hr) (***Figure 6E-H***). However, the combined action of cyclopamine and U0126 caused a rapid downregulation of RUNX2 (24 hr) (***Figure 6E-H***). These findings collectively indicate that pERK1/2 regulates RUNX2 and GLI1 expressions. Whereas, RUNX2 is a predominant downstream target of pERK1/2 and a weak target of GLI1.

## Discussion

The calvarial suture allows us to study many developmental processes that regulate osteogenesis and skull morphogenesis. Normal development is ensured by the correct temporal and spatial coordination of skeletal patterning, mesenchymal stem cell niche regulation, condensation formation, osteoprogenitor expansion, and osteoblast differentiation and function. In this study, we generated *Rab23*-deficient mice that survive into late gestation and they exhibit multiple suture craniosynostosis. We show that RAB23 regulates the sutural stem cell niche marker GLI1, represses Hh, and inhibits osteoprogenitor expansion. RAB23 also represses FGF10-driven pERK1/2 signaling which has the multiple effect of directly regulating osteoprogenitor development, RUNX2-mediated differentiation and of repressing GLI1. Therefore, RAB23 coordinates suture biogenesis by regulating FGF-ERK1/2-RUNX2 activity, as well as the activation of GLI1. In an attempt to study complex developmental events, we have utilized both primary cells and organ culture approaches. However, one can also take a more cell biological approach using immortalized cells remote from the external factors and by using this standardized approach it may be easier to understand the effects of a specific intervention. This will be help deciphering the role of RAB23, specifically during endocytosis.

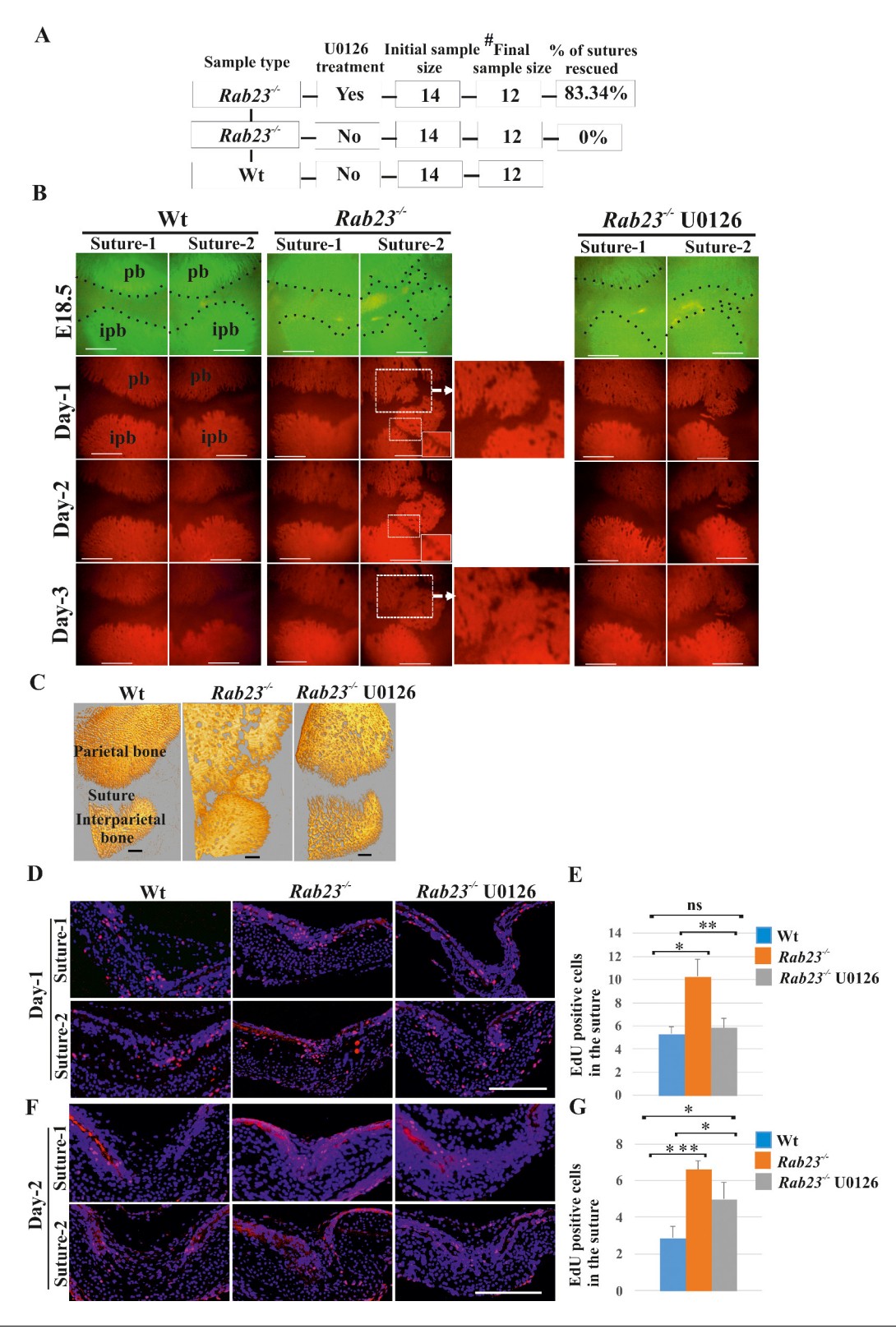

**Figure 5.** Rescuing *Rab23*⁻/⁻ lambdoid suture fusion with pERK1/2 inhibitor. (A–C) In vivo calcein green incorporation followed by in vitro calvaria culture in osteogenic medium and adding alizarin red in medium allowed to follow the bone growth in the sutural site. Layout of lambdoid suture rescue study (A). E18.5 Wt and a double number of *Rab23*⁻/⁻ calvaria were cultured in vitro for 3 day. Half of the *Rab23*⁻/⁻ calvaria were taken for control (DMSO) treatment and the other half were treated with ERK1/2 inhibitor U0126. 100% of *Rab23*⁻/⁻ control sutures are fused by day 3 of culture (A, B, *Figure 5 continued on next page*

*Figure 5 continued*

alizarin red bone staining and C, μ-CT images) while 83.34% of *Rab23⁻/⁻* suture with U0126 treated shows no fusion (**A**, **B**). # indicates dropout of two samples from each group due to technical hindrance. Dotted lines indicate bone edge. pb: parietal bone, ipb: interparietal bone. Scale bar: 500 μm (**B**, **C**). (**D–G**) Represents EdU incorporation in the E18.5 calvaria after 24 hr and 48 hr of culture. EdU was added in the culture medium for additional 2 hr and cell proliferation analyzed in the sectioned Wt, control *Rab23⁻/⁻* and U0126 treated *Rab23⁻/⁻* lambdoid sutural samples (**D**, **F**). U0126 reduced the cell proliferation in *Rab23⁻/⁻* samples compare to untreated *Rab23⁻/⁻* samples at 24 hr and 48 hr. U0126 treated *Rab23⁻/⁻* suture shows dramatic reduction of cell proliferation compare to untreated *Rab23⁻/⁻* suture (**D**, **E**). The reduction of cell proliferation has shown consistent at 48 hr (**F**, **G**) (n = 8 for each genotype). Data represented as mean ± SD, paired Student's *t*-test was used. Statistical significance was defined as a p-value <0.05 (*), p-value <0.005 (**), p-value <0.001 (***). Scale bar: 200 μm (**D**, **F**).

The online version of this article includes the following video and figure supplement(s) for figure 5:

**Figure supplement 1.** Rescuing suture fusion in *Rab23⁻/⁻* calvaria.

**Figure supplement 2.** Rescuing suture fusion in E17.5 *Rab23⁻/⁻* mice.

**Figure supplement 3.** Effect of U0126 on Wt and *Rab23⁻/⁻* lambdoid suture.

**Figure 5—video 1.** Wt lambdoid suture after 72 hr in vitro culture.

https://elifesciences.org/articles/55829#fig5video1

**Figure 5—video 2.** *Rab23⁻/⁻* lambdoid suture fused after 72 hr in vitro culture.

https://elifesciences.org/articles/55829#fig5video2

**Figure 5—video 3.** U0126 treated *Rab23⁻/⁻* lambdoid suture remained open after 72 hr in vitro culture.

https://elifesciences.org/articles/55829#fig5video3

FGF10 signaling regulates early osteogenesis: it is expressed in the osteogenic condensation, and is implicated in the pathogenesis of several craniosynostotic models (*Twigg and Wilkie, 2015*; *Veistinen et al., 2009*). Here, we show that *Fgf10,* its receptor FGFR1b and the FGF activator *Pitx2,* are all upregulated in *Rab23*-deficient calvaria. It is surprising that the FGFR1b splice form, which is normally expressed in epithelia, is upregulated in the mesenchymal suture. However, this is not without precedent. While FGFRc isoforms are predominantly expresses in the mesenchyme and b isoforms in the epithelium, their expression is not mutually exclusive and FGFR1b is known to be expressed by late embryonic murine lung mesenchymal cells and also in the zebrafish in the sutural mesenchyme (*Al Alam et al., 2015*; *Rice et al., 2003b*; *Rice et al., 2004*; *Topczewska et al., 2016*). Under pathological/force conditions, mis-expression of splice forms can occur. Occasionally, Apert craniosynostosis syndrome is caused by heterozygous Alu-insertions or large deletions in the FGFR2c domain. These rare mutations induce abnormal expression of the FGFR2b splice form alongside FGFR2c splice form in mesenchymal cells. This permits cells to respond to both b- and c-activating FGF ligands including FGF10 (*Bochukova et al., 2009*; *Oldridge et al., 1999*). Analyzing the MAPK pathway in *Rab23⁻/⁻* lambdoid sutures we show that p38 signaling is decreased and pERK1/2 increased. FGFR activation of pERK1/2 is important for several stages of osteoblastogenesis, including osteoprogenitor proliferation, and in the induction and stabilization of RUNX2 (*Choi et al., 2008*; *Yoon et al., 2014*). Augmentation of pERK1/2 signaling either through overactivation of *FGFRs*, downregulation of its inhibitors, or downregulation of *ERF* that regulates the export of pERK from the nucleus, all result in craniosynostosis (*Lee et al., 2018a*; *Lee et al., 2018b*; *Timberlake et al., 2017*; *Twigg et al., 2013*).

Although RUNX2 is a substrate for both p38 and pERK1/2, pERK1/2 can activate RUNX2 on average six times more efficiently than p38 (*Ge et al., 2012*). In addition, ERK1/2 binding to RUNX2 through MAPK D site has greater affinity than p38 binding and more sensitive to osteoblast differentiation during calvaria explant or primary osteoblast culture (*Franceschi and Ge, 2017*; *Ge et al., 2012*). The importance of pERK1/2 regulation of RUNX2 during osteogenesis is shown in this study by the reversal of the craniosynostotic phenotype in *Rab23⁻/⁻* mice. p38 has been implicated in the skull phenotypes of craniosynostotic mice where it may regulate the development and growth of cartilage elements (*Wang et al., 2010*; *Wang et al., 2012*). Opposing roles of MAPK pathway subtypes p38 and pERK1/2 in cartilage development are well documented (*Ma et al., 2019*; *Oh et al., 2000*; *Stanton et al., 2003*).

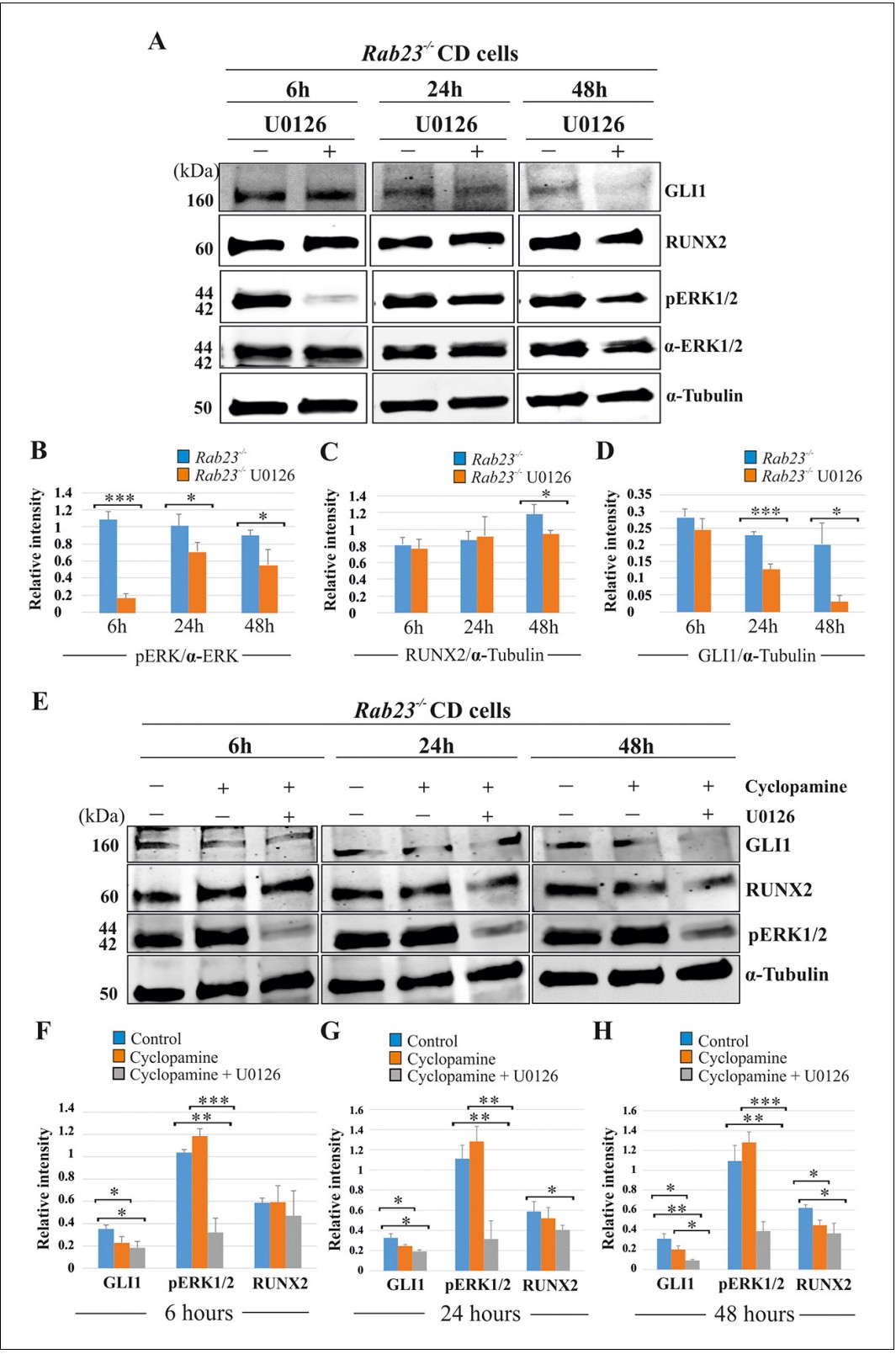

**Figure 6.** RUNX2 and GLI1 are downstream target of pERK1/2. (**A–D**) Time-dependent effect of U0126 in regulating pERK1/2, RUNX2 and GLI1 expressions has been analyzed by western blotting in the proteins extracted from E15.5 *Rab23−/−* CD cells. These cells were treated with or without U0126. Relative pERK1/2 levels in *Rab23−/−* CD cells are drastically reduces upon U0126 exposure at 6 hr and also shows downregulation of pERK1/2 at 24

*Figure 6 continued on next page*

*Figure 6 continued*

and 48 hr upon U0126 exposure compare to corresponding untreated *Rab23⁻/⁻* CD cells (**A, B**). RUNX2 expression significantly reduces in *Rab23⁻/⁻* CD cells only after 48 hr of U0126 exposures (**A, C**). GLI1 expression in *Rab23⁻/⁻* CD cells sequentially downregulated upon U0126 exposures at 6, 24 and 48 hr (**A, D**) (n = 3 blots for each time points). (**E–H**) Time-dependent effect of cyclopamine (10 µM) and combined effect of cyclopamine (10 µM) and U0126 (5 µM) in regulating GLI1, pERK1/2, and RUNX2 expressions has been analyzed by western blotting in the proteins extracted from E15.5 *Rab23⁻/⁻* CD cells. These cells were treated with or without cyclopamine and combined cyclopamine and U0126 for 6, 24 and 48 hr. Relative GLI1 levels are significantly reduces at every time points upon cyclopamine treatment and show further gradual reduction of GLI1 upon combined treatment of cyclopamine with U0126 (**E–H**). Relative pERK1/2 levels in *Rab23⁻/⁻* CD cells are drastically reduces upon combined cyclopamine and U0126 exposure at 6 hr (**E, F**) and also shows downregulation of pERK1/2 at 24 and 48 hr upon combined cyclopamine and U0126 exposure compare to corresponding untreated *Rab23⁻/⁻* CD cells (**E, G, H**). However, pERK1/2 level remain unchanged at every time points upon cyclopamine treatment alone. RUNX2 expression significantly reduces in *Rab23⁻/⁻* CD cells after 24 hr by combined treatment of cyclopamine and U0126 (**E, G, H**), and only significantly reduces after 48 hr upon cyclopmine treatment alone. (n = 3 blots for each time points). Data represented as mean ± SD, paired Student's *t*-test was used. Statistical significance was defined as a p-value <0.05 (*), p-value <0.005 (**), p-value <0.001 (***).

The online version of this article includes the following figure supplement(s) for figure 6:

**Figure supplement 1.** GLI1 is a downstream target of pERK1/2 Immunohistochemical staining shows GLI1 expression in *Rab23⁻/⁻* and U0126-treated *Rab23⁻/⁻* lambdoid sutural sections at 24 and 48 hr of in vitro culture of E18.5 samples.

---

Commonality of etiological mechanisms is suggested by the overlapping skeletal phenotypes exhibited in Carpenter syndrome and Grieg cephalopolysyndactyly syndrome patients as well as their mouse disease models, *Gli3^(Xt−J/Xt−J)* (*Rice et al., 2010*) and *Rab23⁻/⁻* (this study). Specifically, postaxial polydactyly in the forelimb, preaxial polydactyly in the hindlimb and lambdoid and inter-frontal suture craniosynostosis (*Eggenschwiler et al., 2001*; *Jenkins et al., 2007*; *Kalff-Suske et al., 1999*; *Vortkamp et al., 1991*). Collectively suggest that craniosynostosis and polysyndactyly are overlapping phenotypes of aberrant FGF and Hh signaling. Aberrant osteogenesis and suture obliteration in the *Gli3^(Xt−J/Xt−J)* mouse is caused by concomitant derepression of GLI3 and augmentation of IHH which activates the osteogenic master regulatory transcription factor RUNX2 (*Veistinen et al., 2017*). Analysis of early *Rab23⁻/⁻* embryos suggests that RAB23 also has a role in promoting the production of GLI3 repressor (*Eggenschwiler et al., 2006*). In both *Gli3^(Xt−J/Xt−J)* and *Rab23⁻/⁻* calvaria elevated Hh signaling results in an upregulation of the transcription factor Gli1, and presumably the mesenchymal stem niche which it labels.

Previously, we have shown that FGF10 upregulates *Hh* and its receptor *Patch1* during facial development (*Rice et al., 2004*). Here, we show FGF10 upregulates Runx2. Our study revealed that inhibition of pERK1/2 prevented *Rab23⁻/⁻* lambdoid suture fusion by decreasing osteoprogenitor proliferation and GLI1 expression (*Figures 5* and *6*). We further showed that combined and individual inhibition of GLI1 and pERK1/2 in *Rab23⁻/⁻* calvarial cells downregulated RUNX2 and GLI (*Figure 6*) and that this effect was exaggerated in *Rab23⁻/⁻* cells. Phenotypic analyses of compound *Rab23;Smoothened* early embryonic mice shows that RAB23 can regulate GLI independently of Smoothened (*Eggenschwiler et al., 2006*), and the regulation of GLI1 by pERK1/2 suggests a mechanism how this might occur (*Figure 7*). Non-canonical regulation GLI1 through tyrosine kinase activity has been documented during carcinogenesis, with activation of GLI1 promoting stemness through an increase in transcriptional activity, nuclear localization or protein stability (*Pietrobono et al., 2019*). However, the non-canonical repression of GLI1 has not been well described in a developmental context.

We show that RAB23 represses both FGF and Hh/GLI signaling (*Figure 7*). RAB23 coordinates suture morphogenesis and controls suture patency through three mechanisms: FGF10-pERK1/2 signaling, the repression of Hh targets through a canonical pathway, and through the

repression of GLI1 by pERK1/2. GLI1 contributes in the regulation of the mesenchymal stem cell niche. FGF10-pERK1/2 and also Hh-GLI1 signaling regulate osteoprogenitor cell proliferation.

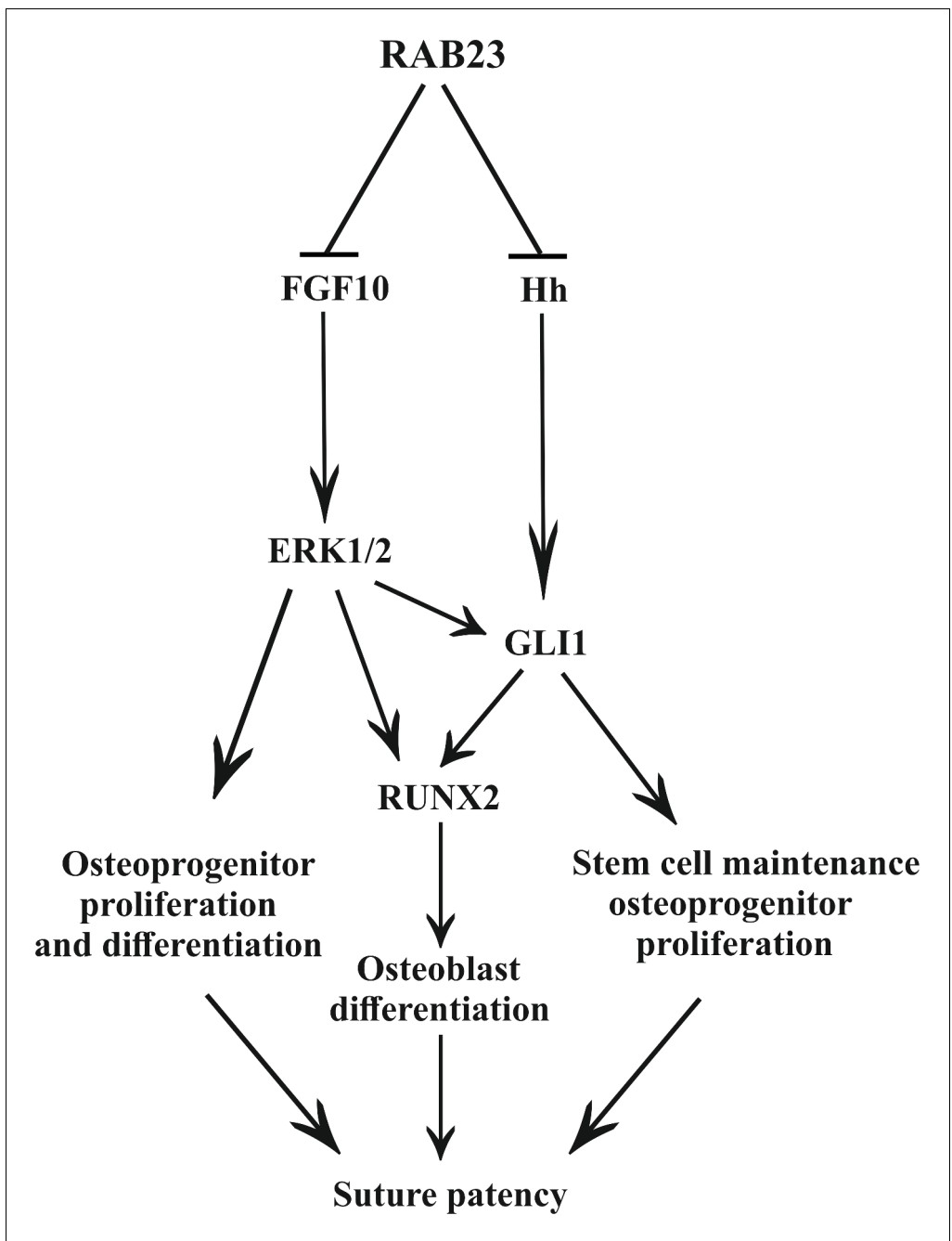

**Figure 7.** Model: regulation of suture patency by RAB23. In the developing calvaria RAB23 regulates FGFR signaling by repressing FGF10 expression. This regulates the delicate balance of osteoprogenitor proliferation and differentiation. RAB23 also known as a negative regulator of Hh signaling. The regulation of GLI1 through ERK1/2 and Hh in the mesenchymal stem cell niche is maintained along with RUNX2 during osteoprogenitor proliferation and differentiation. Thus, through a combination of FGF and Hh signaling, RAB23 tightly synchronizes suture morphogenesis and patency.

# Materials and methods

## Key resources table

| Reagent type (species) or resource | Designation | Source or reference | Identifiers | Additional information |
|---|---|---|---|---|
| Strain, strain background (*M. musculus*, C3Heb/FeJ) | *Rab23opb2* mice C3Heb/FeJ | *Eggenschwiler et al., 2001*; *Kasarskis et al., 1998* | | |
| Strain, strain background (*M. musculus*, C57/Bl6) | *Rab23opb2* mice C57Bl/6 | This paper | | |
| Cell lines (*Mus musculus*) | | This paper | | |
| Antibody | Rabbit polyclonal RAB23 | Proteintech | Cat#11101–1-AP | WB: 1:1000, IHC: 1:250 |
| Antibody | Rabbit monoclonal FGFR1 | Cell signaling technology | Cat#9740 | WB: 1:1000, IHC: 1:250 |
| Antibody | Rabbit monoclonal p44/42 | Cell signaling technology | Cat#4965S | WB: 1:1000 |
| Antibody | Rabbit monoclonal phospho-p44/42 | Cell signaling technology | Cat#9101 | WB: 1:1000 |
| Antibody | Rabbit monoclonal phospho-p38 | Cell signaling technology | Cat#4631S | WB: 1:1000 |
| Antibody | Rabbit polyclonal p38 | Cell signaling technology | Cat#9212 | WB: 1:1000 |
| Antibody | Rabbit monoclonal RUNX2 | Cell signaling technology | Cat#8486 | WB: 1:1000, IHC: 1:250 |
| Antibody | Mouse monoclonal GLI1 | Cell signaling technology | Cat#2643 | WB: 1:1000, IHC: 1:250 |
| Antibody | Rabbit monoclonal pSMAD2/3 | Cell signaling technology | Cat#8828 | WB: 1:1000 |
| Antibody | Rabbit polyclonal pSMAD1/5/8 | Millipore | Cat#AB3848 | WB: 1:1000 |
| Antibody | Mouse monoclonal RUNX2 | Santa Cruz Biotechnology | Cat# sc-390351 | WB: 1:1000, IHC: 1:250 |
| Antibody | Mouse αTubulin | Sigma-Aldrich | Cat#T6199 | 1:2000 |
| Antibody | Goat anti-rabbit IgG (H+L), Alexa 488 | Thermo Fisher Scientific | Cat#A-11008 | 1:500 |
| Antibody | Goat anti-mouse IgG (H+L), Alexa 546 | Thermo Fisher Scientific | Cat#A-110003 | 1:500 |
| Antibody | Goat anti-mouse IgG (H+L), Alexa 488 | Thermo Fisher Scientific | Cat#A-11001 | 1:500 |
| Antibody | Goat anti-rabbit 680LT | LI-COR | Cat#925–68021 | 1:5000 |

*Continued on next page*

*Continued*

| Reagent type (species) or resource | Designation | Source or reference | Identifiers | Additional information |
|---|---|---|---|---|
| Antibody | Goat anti-rabbit 800CW | LI-COR | Cat#925–32211 | 1:5000 |
| Antibody | Goat anti-mouse IRDye 800CW | LI-COR | Cat#925–32210 | 1:5000 |
| Commercial assay or kit | Enzmet HRP detection kit | Nanoprobes | Cat#6001 | |
| Commercial assay or kit | Hoechst 33342 | Thermo Fisher Scientific | Cat# H3570 | 1:2000 |
| Commercial assay or kit | Pierce IP lysis buffer | Thermo Fisher Scientific | Cat#87787 | |
| Commercial assay or kit | RevertAid reverse transcriptase | Thermo Fisher Scientific | Cat#EP0441 | |
| Commercial assay or kit | Random hexamer primer | Thermo Fisher Scientific | Cat#SO142 | |
| Commercial assay or kit | Ribolock RNase inhibitor | Thermo Fisher Scientific | Cat#EO0381 | |
| Commercial assay or kit | Odyssey blocking buffer | LI-COR | 927–40100 | |
| Commercial assay or kit | FGF10 | R and D | Cat#6224-FG-025 | |
| Commercial assay or kit | Complete protease inhibitor cocktail | Roche | Cat#04693116001 | |
| Commercial assay or kit | Complete phosphatase inhibitor cocktail | Roche | Cat#04906845001 | |
| Commercial assay or kit | Nucleospin RNA | Biotop | Cat#740955.250 | |
| Commercial assay or kit | Click-iT EdU Alexa Fluor 594 | Invitrogen | Cat#C10339 | |
| Commercial assay or kit | EdU (5-ethynyl-2′-deoxyuridine) | Sigma-Aldrich | Cat#A10044 | |
| Commercial assay or kit | U0126 | Sigma-Aldrich | Cat#662005 | |
| Commercial assay or kit | Alizarin-3-methyliminodiacetic acid | Sigma-Aldrich | Cat#A3882 | |
| Commercial assay or kit | Calcein green | Sigma-Aldrich | Cat#C0875 | |
| Commercial assay or kit | Apoptosis detection kit | Abcam | Cat#ab206386 | |
| Commercial assay or kit | Pierce BCA protein assay kit | Thermo Fisher Scientific | Cat#23225 | |
| Commercial assay or kit | KAPA SYBR FAST qPCR master mix (2X) | KAPA Biosystems | Cat#KK4601 | |
| Commercial assay or kit | 4–20% Mini-PROTEAN TGX Gels | BIORAD | Cat#456–1094 | |
| Sequence-based reagent | *Pitx2* | This study | NM_001286942.1 | *F: CCGCCTGGCAGTCACC* *R: CTCCATTCCCGGTTATCGGC* |
| Sequence-based reagent | *Fgf10* | This study | NM_008002.4 | F: AGGCTGTTCTCCTTCACCAAG R:ATGTTATCTCCAGGACACTGTACG |
| Sequence-based reagent | *Gli1* | This study | NM_010296.2 | *F: CAGCATGGGAACAGAAGGACT* *R: CTCTGGCTGCTCCATAACCC* |

*Continued on next page*

Continued

| Reagent type (species) or resource | Designation | Source or reference | Identifiers | Additional information |
|---|---|---|---|---|
| Sequence-based reagent | *Gli2* | This study | NM_001081125.1 | *F: AACTTTTGTCTCCTCGGGTCC R: CTGCTGTCCTCCAAGAGACC* |
| Sequence-based reagent | *Gli3* | This study | NM_008130.3 | *F: AAGCCCATGACATCTCAGCC R: CTCGAGCCCACTGTTGGAAT* |
| Sequence-based reagent | *Hh* | This study | XM_006535649.4 | *F: AAGCAGGTTTCGACTGGGTC R: CCACGGAGTTCTCTGCTTTCA* |
| Sequence-based reagent | *Fgfr1b* | *Lee et al., 2008* | | *F: GGGAATTAATAGCTCGGATGA R: ACGCAGACTGGTTAGCTTCA* |
| Sequence-based reagent | *Fgfr2b* | This study | NM_201601.2 | *F: TCAAGGTCCTGAAGCACTCG R: CAGCATCCATCTCCGTCACA* |
| Sequence-based reagent | *Fgfr2c* | This study | NM_010207.2 | *F: AACCAGAAGAGCCACCAACC R: TAGTCCAACTGATCACGGCG* |
| Sequence-based reagent | *Runx2* | This study | NM_001271631.1 | *F: CAGTCCCAACTTCCTGTGCT R: CCCATCTGGTACCTCTCCGA* |
| Sequence-based reagent | *18S rRNA* | *Nishioka et al., 2010* | | *F: AAACGGCTACCACATCCAAG R: CAATTACAGGGCCTCGAAAG* |
| Software | Fiji (Image J) | National Institute of Health | https://fiji.sc/ | |
| Software | Desktop micro-CT system | Bruker | SkyScan1272 | |
| Software | Nrecon, Desktop micro-CT system | Bruker | SkyScan1272 | |
| Software | Odyssey infrared imaging system | LI-COR Biosciences | Model 9120 | |

## Study approval

Animal experiments were approved by the University of Helsinki, Helsinki University Hospital and the Southern Finland Council Animal Welfare and Ethics committees. Permit ESAVI/11956/04.10.07/2017 (external) and KEK17-008 (internal).

## Mice

*Rab23*$^{opb2/opb2}$ (*Rab23*$^{-/-}$) mice were originally in C3Heb/Fej background were crossed into C57Bl/6J mouse strain (Charles River) for six or more generations with speed congenic method. The animals for breeding were selectively based on highest amount of C57 SNP markers (Illumina mouse chip array) to speed up strain change. After four generations more than 89% of SNPs were on C57. The genotype of all mouse and embryos were verified with PCR-based genotyping as previously described (*Eggenschwiler and Anderson, 2000*; *Kasarskis et al., 1998*).

## Microarray and RT-qPCR

Analysis of the RNA expression was performed using Illuminan Mouse WG-6v2 microchips. RNA was isolated from E15 calvaria from *Rab23*$^{-/-}$ embryos and Wt littermates using Machrey-Nagel Nucleo-spin RNA II isolation kit. Labeling and chip run were performed by the Technology Centre, Institute for Molecular Medicine Finland (FIMM), University of Helsinki. The data analysis and normalization was performed using Chipster 3.16.0. Fold change of genes is calculated by arithmetic mean in linear scale and shown in the volcano plot. Fold change >1 (up-regulated gene), fold change < 1 (down-regulated gene). The microarray dataset has been deposited in MIAME-compliant GEO public database as source data. Accession number: GEO accession GSE140884. For quantitative PCR, RNA was isolated individually from whole calvaria and from lambdoid suture of *Rab23*$^{-/-}$ and Wt embryos at E15.5. 1 μg of RNA from calvarial (n = 3+3) and 500 ng of RNA from lambdoid suture

(n = 8+8) were reverse-transcribed into cDNA using random hexamer primer and RevertAid reverse transcriptase (Thermo scientific). 2 µl of diluted cDNA and Brilliant III ultra-fast SYBR-green qPCR master mix with QuantiTect Primers (Qiagen).

## RNAscope in situ hybridization

We performed the experiments as previously described (*Sanz-Navarro et al., 2019*). In single-gene in situ RNAscope transcripts appear in red. A customized 20ZZ probe named Mm-*Rab23* targeting 45–1330 of NM_008999.4 was purchased from Advanced Cell Diagnostics to perform *Rab23* expression analysis on mouse calvarial suture. Images were obtained using a bright field microscope (BX61, Olympus).

## In situ hybridization on whole mounts

In situ hybridizations on whole embryonic samples was performed using digoxigenin-UTP labeled *Rab23* probe and *Pitx2* probe according to the protocol and modification described (*Kim et al., 1998*).

## Immunohistochemistry

Immunohistochemistry on lambdoid sutural samples was performed using primary anti-RAB23, GLI1, RUNX2 and FGFR1 antibody. After incubation with HRP conjugated or alexa fluor conjugated secondary antibody Enzmet detection kit was used for GLI1 and FGFR1 detection and alexa fluor conjugated 488, 546 secondary antibody was used to detect RAB23 and RUNX2 signals. The samples were permeabilized by triton X-100 and antigen retrieval was performed by citrate buffer. All the other procedure was followed according to the protocol described (*Veistinen et al., 2017*).

## Isolation and culture of mouse calvaria derived primary cells

WT and *Rab23*$^{-/-}$ embryos were taken out at E15.5 embryonic day. We and others have previously described the procedure of calvarial primary cells isolation (*Rodan and Noda, 1991*; *Veistinen et al., 2017*). Cells were cultured in DMEM containing high glucose and supplemented with 10% FBS, glutamine, penicillin and streptomycin, maintained at 37°C with 5% $CO_2$. After maintaining the isolated cells in growth medium next passage is always used for experiments.

## Exogenous FGF10 treatment on calvaria derived mesenchymal cells

E15.5 Wt calvaria derived mesenchymal cells were treated with exogenous FGF10 (250 ng/ml) for 3 hr. *Fgfr1b* and *Fgfr2b* mRNA expression were then assessed by RT-qPCR. E15.5 *Rab23*$^{-/-}$ calvaria derived mesenchymal cells were treated with exogenous FGF10 (500 ng/ml) for 2 and 4 hr. *Runx2* mRNA expression were then assessed by RT-qPCR.

## Tissue lysis, protein extraction, quantification and western blotting

RIPA buffer containing 1% SDS together with protease inhibitor and phosphatase inhibitor were used to lyse the lambdoid sutural tissue on ice. After brief sonication (10 s, twice), tissue lysates were centrifuged at full speed for 10 min to collect the protein supernatant. By using BCA protein assay kit, protein concentrations were measured for immunoblotting analysis. Equal amount of proteins from wild type and RAB23 deficient cells were subjected to prepare to separate on SDS-PAGE under reduced gel electrophoresis (4–20% Mini-PROTEAN TGX Gels, Bio-Rad). After transferring to nitrocellulose membrane, membranes were blocked by odyssey blocking buffer at room temperature for 3 hr. Membranes were then incubated with corresponding primary anti-RAB23, anti-p38, anti-phospho-p38, anti-ERK, anti-pERK, anti-RUNX2, anti-pSMAD1/5/8, anti-GLI1, anti-FGFR1 and anti-Tubulin antibodies and kept overnight at 4°C. Fluorophore-conjugated corresponding IRdye goat anti-rabbit 680LT or CW800 or anti-mouse 800CW secondary antibodies were used at room temperature for 1 hr. α-Tubulin was used to normalize protein expressions. In vitro inhibition of pERK1/2 in *Rab23*$^{-/-}$ calvaria derived primary cells (E15.5) was performed by adding 5 µM of U0126 and 10 µM of cyclopamine for 6, 24 and 48 hr. Membranes were scanned using an Odyssey infrared imaging system (Odyssey; LI-COR Biosciences, model 9120). Band intensity was determined using ImageJ (NIH) software.

## Craniosynostosis rescue study and microscopy

After removing skin and underneath brain calvaria from E18.5 Wt and *Rab23*[-/-] were cultured in an in vitro explant culture system. The calvaria was kept on top of the filter laid on metallic grid. The calvaria sample was supplemented with DMEM containing 10% FBS, dexamethasone, ascorbic acid and β-glyceraldehyde. *Rab23*[-/-] calvaria received 5 µM of U0126 treatment or DMSO as control. After 48 hr fresh medium and inhibitors were added. After 48 or 72 hr, samples were fixed with 4% PFA for subsequent analysis. For immediate calvarial bone and suture recognition vital bone dye calcein green was injected to mother mouse 1 day before the E17.5 or E18.5 embryo collection. Subsequent imaging in the culture was performed after alizarin red incorporation in the medium. Images were taken by Olympus SZX12 fluorescent microscope.

## Skeletal staining

The Alcian blue-Alizarin red staining of E18.5 specimens was performed as previously described (*Rice et al., 2003a*). For analysis of calvarial bone and sutures images were captured using Analysis software (Soft Imaging System) and Olympus BX41 microscope and analyzed in Adobe Photoshop CS4.

## MEF isolation

Mouse embryonic fibroblasts (MEFs) were isolated from E13.5 embryos (noon of the vaginal plug detection day was considered E0.5). Briefly, embryos were collected to sterile PBS. The head, liver and spleen were removed and used for histology, RNA isolation or for genotyping. The remaining tissue was cut into smaller pieces and dissociated with gentle pipetting in 0.25% Trypsin-EDTA solution. After 10 min incubation, any larger tissue pieces were discarded and remaining cell suspension was plated with rich MEF medium into two T25 flasks per embryo. The medium was changed daily until first passaging at confluency.

The growth medium was DMEM, High Glucose, GlutaMAX (Gibco) supplemented with 10% FBS, 1x non-essential amino acids (Gibco), 20 mM Hepes pH 7.3 and 1x penicillin-streptomycin. The cells were subcultured when 90% confluent to 750,000 live cells/T75 flask and only passages 1–6 (active division) were used for EdU pulsed proliferation analyses. The cell density and viability was assessed with Trypan Blue and Countess automated cell counter (Invitrogen).

## EdU pulsing and staining

Cell proliferation was assessed by prior EdU incorporation followed by EdU click reaction. The MEF cell lines were pulsed for 1 hr with 10 µM EdU (Molecular probes, Invitrogen) in DMSO dissolved to pre-warmed medium before distribution to cells for 1 hr EdU pulse and fixed with 4% paraformaldehyde. The EdU was detected with Click-iT EdU imaging kit, Alexa fluor 594 (Molecular probes, Invitrogen) according to manufacturer's instruction. The number of EdU positive cells was counted from three separate experiments from a minimum five different locations per cell line and compared to total cell number determined with Hoechst 33342 staining of nuclei.

The mice were pulsed with 0.05 mg/g (i.p.) EdU in PBS 2 hr collection of tissue. After sacrificing E17.5 embryonic lambdoid sutures were collected and fixed o/n with 4% paraformaldehyde, and processed for paraffin sections. Paraffin-embedded lambdoid sutures were sectioned 7 µm size. The sections were deparaffinized and stained with same Click-iT kit. EdU click reaction procedure in brief, sectional explants were deparaffinized by xylene and rehydrated by a gradient of ethanol series. After washing with 2 mg/ml glycine, sutural sections were permeabilized with 0.5% Triton X-100. After several washing in PBS, 10 µM EdU cocktail was used for click reaction for 30 min at dark room. After several PBS washing 5 µg/ml Hoechst was used for counter staining. Vectashield mounting medium was used to mount the slides. Slides were then imaged with fluorescence microscope with emission wavelength 615 nm.

Cell counting: Cluster of proliferating cells (red cell) considered as osteogenic front. The middle of the scale was kept on the tip of each osteogenic front and then rest of the part was flanked towards the suture. The length of the scale is 145 µm and width 61 µm. Cells from skin, dura mater and cartilage were not included in counting.

## Apoptotic assay

Apoptotic assay on tissue sections from in vitro cultured Wt, control $Rab23^{-/-}$ and U0126 treated $Rab23^{-/-}$ lambdoid suture was performed according to manufacturer protocol (In situ apoptosis detection kit AB206386).

## X-ray micro computed tomography

E18.5 WT and $Rab23^{-/-}$ whole head and samples from rescue studies were collected and fixed with 4% paraformaldehyde overnight. Samples were dehydrated by gradient ethanol series to 70% (v/v), followed by processed for x-ray microtomography (μCT) imaging with Bruker SkyScan1272 (desktop micro-CT system, Bruker microCT N.V., Kontich, Belgium). Tomography 3D reconstructions were obtained using the program NRecon (desktop micro-CT system, Bruker microCT N.V., Kontich, Belgium).

## Data analysis and statistics

Paired student's *t-test* has been applied to perform the statistics of all the data obtained from measurements, western blotting and gene expressions. Data represented as mean ± SD. p-Value less than 0.05 considered as statistical significant.

## Acknowledgements

We thank Jonathan Eggenschwiler for providing us $Rab23^{opb2}$ mouse, Pekka Nieminen for his critical and constructive discussion on this project. We also thank Airi Sinkko and Anne Kivimäki for their excellent technical help.

## Additional information

### Funding

| Funder | Grant reference number | Author |
|---|---|---|
| Suomen Akatemia | 257472 | David PC Rice |
| Sigrid Juséliuksen Säätiö | 4702957 | David PC Rice |
| Helsinki University Hospital Research Foundation | TYH2019250 | David PC Rice |

The funders supported to perform this research.

### Author contributions

Md Rakibul Hasan, Conceptualization, Resources, Data curation, Software, Formal analysis, Validation, Investigation, Visualization, Methodology, Writing - original draft, Writing - review and editing; Maarit Takatalo, Conceptualization, Resources, Data curation, Formal analysis, Investigation, Visualization, Methodology, Writing - original draft, Writing - review and editing; Hongqiang Ma, Conceptualization, Resources, Software, Formal analysis, Investigation, Visualization, Writing - original draft, Writing - review and editing; Ritva Rice, Conceptualization, Formal analysis, Investigation, Visualization, Writing - original draft, Writing - review and editing; Tuija Mustonen, Conceptualization, Resources, Formal analysis, Validation, Investigation, Visualization, Writing - original draft, Writing - review and editing; David PC Rice, Conceptualization, Resources, Data curation, Supervision, Funding acquisition, Visualization, Writing - original draft, Project administration, Writing - review and editing

### Author ORCIDs

Md Rakibul Hasan (ID) https://orcid.org/0000-0002-5477-6934
Tuija Mustonen (ID) http://orcid.org/0000-0002-2429-5064
David PC Rice (ID) https://orcid.org/0000-0001-9301-3078

## Ethics

Animal experimentation: Animal experiments were approved by the University of Helsinki, Helsinki University Hospital and the Southern Finland Council Animal Welfare and Ethics committees. Permit ESAVI/11956/04.10.07/2017 (external) and KEK17-008 (internal).

## Decision letter and Author response

Decision letter https://doi.org/10.7554/eLife.55829.sa1
Author response https://doi.org/10.7554/eLife.55829.sa2

## Additional files

### Supplementary files
- Supplementary file 1. List of 223 differentially expressed genes.
- Transparent reporting form

### Data availability

MIAME-compliant microarray data has been deposited in GEO database. GEO accession GSE140884 The dataset link: https://www.ncbi.nlm.nih.gov/geo/query/acc.cgi?acc=GSE140884.

The following dataset was generated:

| Author(s) | Year | Dataset title | Dataset URL | Database and Identifier |
|---|---|---|---|---|
| Hasan MR, Takatalo M, Rice DP | 2020 | Microarray based gene expression analysis in Wt and Rab23-/- mice calvaria at embryonic day 15.5 | https://www.ncbi.nlm.nih.gov/geo/query/acc.cgi?acc=GSE140884 | NCBI Gene Expression Omnibus, GSE140884 |

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
