## [Decision Letter]

**Acceptance summary:**

The authors have effectively determined the mechanism underlying craniosynostosis caused by *Rab23* deletion and determined the function of RAB23 during intramembranous bone formation. Notably, RAB23 coordinates skull morphogenesis by regulating suture patency through FGF10-ERK1/2RUNX2 signaling, repressing *Hh* targets through a canonical pathway, and regulating the sutural stem cell marker GLI1.

**Decision letter after peer review:**

Thank you for submitting your article "RAB23 coordinates early osteogenesis by repressing FGF10-pERK1/2-RUNX2 and GLI1" for consideration by *eLife*. Your article has been reviewed by three peer reviewers, and the evaluation has been overseen by Mone Zaidi as the Reviewing Editor and Clifford Rosen as the Senior Editor. The following individual involved in review of your submission has agreed to reveal their identity: Nan Hatch (Reviewer #1).

The reviewers have discussed the reviews with one another, and on basis of their recommendation, we are requesting a revised submission. In recognition of the fact that revisions may take longer than the two months we typically allow due to the COVID-19 pandemic, until the research enterprise restarts in full, we will give authors as much time as they need to submit revised manuscripts.

Summary:

The authors have generated and studied global *Rab23*^-/-^ mice at embryonic time points to establish an essential role for *Rab23* in skull growth and morphogenesis. These mice serve as a useful model for Carpenter Syndrome, features of which include craniosyntososis and dactyly. The current work address mechanism by integrating studies of developmental anatomy, gene expression, analysis of cell proliferation and pharmacological inhibition. While there was overall interest in the manuscript and its conclusions, significant concerns were raised regarding the approach, specifically arising from the absence of conclusive experiments.

Essential revisions:

1) The authors do not take a cell biological approach to dissect the primary chain of events leading to the phenotypes they observe. Furthermore, the experimental results are based on collections of what are undoubtedly highly heterogeneous cell types, namely whole calvaria, cultured calvaria-derived mesenchymal cells, dissected and cultured lambdoid sutures, making it impossible to distinguish primary from secondary events in the chain of pathophysiology. This aspect needs clarification.

2) Stronger rationale should be provided earlier in the manuscript explaining why FGF10 and PITX2 were the genes chosen to study the differentially expressed genes. The same could be said for *Hh* and GLI1 expression. The manuscript would benefit from cluing in the reader earlier on regarding rationale for studying these specific factors in their study. Additionally, there is review concern that there is no other evidence for the direct regulation of these molecules by RAB23, and given the caveats above (point 1), secondary mechanisms for these effects are entirely possible. This aspect needs clarification. Furthermore, there is a paucity of robust evidence of the functional significance of any of these molecules in suture fusion. Changed expression levels of any given gene does not necessarily establish its role in a phenotype. The authors should examine in loss- or gain-of-function experiments the functionality of what they consider as the most critical disease-causing gene or gene set.

3) The authors checked premature suture fusion either by CT scanning or alizarin red-alzarin blue staining. Histological analysis, or at least coronal/sagittal sections of the suture CT images, are necessary to confirm the suture fusion. Furthermore, they need to co-stain markers, such as ALP and RUNX2, with their RNAscope analyses (e.g. Figure 1P, middle panel) and immunofluorescence staining images. It is currently difficult to differentiate the suture mesenchyme from the bone at these embryonic stages, which makes data interpretation difficult. Finally, to provide evidence for their conclusion that *Rab23* lambdoid suture and MEF cells show increased osteoprogenitor cell proliferation, the authors must to co-localize Edu with osteoprogenitor cell markers in vivo, noting that Edu will label all proliferating cells.

4) There is no supporting data for a downstream action of FGF10 although both FGF10 and PITX2 are increased in *Rab23*^-/-^ mice. The authors have only validated the in situ expression of *Pitx2*, not of *Fgf10*. There is published data showing that PITX2 is an upstream regulator of FGF10 during early organogenesis of other tissues. Data showing that FGF10 is downstream of PITX2 in suture development is therefore required to strengthen the conclusion.

5) The authors noted an elevated expression of FGF10 and FGFR1b in *Rab23*^-/-^ mice, and concluded that increased FGFR1b was induced by FGF10. There is however the possibility that increased FGFR1b is a direct result of the *Rab23* mutation. This possibility needs to be discussed or excluded experimentally.

6) The authors found elevated pERK1/2, RUNX2, *Hh* and GLI1, as well as decreased p38 in *Rab23*^-/-^ mice. It however remains unclear whether the elevated pERK1/2 and pp38 are secondary to FGF signaling, and whether *Hh* and GLI1 upregulation is caused by phosphorylation of ERK1/2. Interestingly, published data shows that RAB23 can negatively regulate the *Hh* pathway directly. Therefore, it is critical to experimentally establish a direct role for RAB23 in changing gene expression. Promoter-reporter assays could be useful in this regard.

7) The authors used pSmad1/5/8 to test whether the TGF-β superfamily was affected by the *Rab23* mutation. Since pSmad1/5/8 is the readout of global BMP signaling, another readout of the TGF signaling pathway, pSmad2/3, should be tested.

8) It is surprising that the epithelial, but not mesenchymal splice forms of FGFR1 are upregulated in suture and/or calvarial tissue. The authors should make some attempt to explain as to how or why this is the case if the sample should only contain mesenchymal cells.

---

## [Author Response]

Essential revisions:1) The authors do not take a cell biological approach to dissect the primary chain of events leading to the phenotypes they observe. Furthermore, the experimental results are based on collections of what are undoubtedly highly heterogeneous cell types, namely whole calvaria, cultured calvaria-derived mesenchymal cells, dissected and cultured lambdoid sutures, making it impossible to distinguish primary from secondary events in the chain of pathophysiology. This aspect needs clarification.

We thank the reviewers for bringing up this point. We agree that to understand the primary role of RAB23 direct cell biological investigations are needed. Although the direct cellular roles of most RABs are known, very little is understood about RAB23, i.e., role in endocytosis, vesicle formation, recycling and degradation. We agree that by using immortalized cells one can use a standardized approach in which it is easier to understand the effects of a specific intervention. We have added text to the manuscript to acknowledge and clarify these aspects.

It is true that suture morphogenesis and aberrant suture fusion can be investigated by cell biological approach. However, these developmental events are complex and a multiple-pronged strategy is also useful.

By not using such an approach you may bypass the real-time developmental events where multiple regulatory signaling pathways are highly orchestrated. These events regulate not only cell differentiation and function but also regulate tissue-tissue interactions, morphogenesis, patterning, boundary formation within the developing calvaria (Helms et al., 2005) (Twigg and Wilkie, 2015).

As the reviewers mentioned in this study we have combined many approaches (analysis of whole calvaria, cultured calvaria-derived mesenchymal cells, in vitro culture of dissected lambdoid sutures). It is true that some cellular heterogeneity may exist in the calvaria-derived mesenchymal cell cultures but during processing and after enzymatic digestion, epithelial cells, meningeal cells and cartilage cells are lost. Indeed this is a very well investigated, standard technique to harvest osteoblast progenitors (Lana-Elola et al., 2007; Liu et al., 2019; Malaval et al., 1999; Rodan and Noda, 1991)

We believe that by analyzing intact tissues and isolated cells we have provided a broad view of suture fusion in which location-specific information is maintained. Nevertheless, we do acknowledge that further analysis using a more cell biological approach will help in deciphering the role of RAB23, specifically during endocytosis.

2) Stronger rationale should be provided earlier in the manuscript explaining why FGF10 and PITX2 were the genes chosen to study the differentially expressed genes. The same could be said for Hh and GLI1 expression. The manuscript would benefit from cluing in the reader earlier on regarding rationale for studying these specific factors in their study. Additionally, there is review concern that there is no other evidence for the direct regulation of these molecules by RAB23, and given the caveats above (point 1), secondary mechanisms for these effects are entirely possible. This aspect needs clarification. Furthermore, there is a paucity of robust evidence of the functional significance of any of these molecules in suture fusion. Changed expression levels of any given gene does not necessarily establish its role in a phenotype. The authors should examine in loss- or gain-of-function experiments the functionality of what they consider as the most critical disease-causing gene or gene set.

We thank the reviewers for these comments. Stronger rationale for why FGF signaling in particular, *Fgf10* and *Pitx2* were chosen to study further is now given early in the manuscript both in the Introduction section and in the Results section.

Most forms of syndromic craniosynostoses are caused by gain-of-function mutations in FGFR where the receptors exhibit ligand independent activation or lost the ligand binding specificity and bind with cognate ligand (Hajihosseini et al., 2009; Ibrahimi et al., 2001; Ibrahimi et al., 2004; Johnson and Wilkie, 2011; Yu et al., 2000). FGFRs isoforms show distinct FGF ligand binding affinity and biological functions. However, studies have shown that in Apert syndrome (FGFR2 mutation) mouse model FGFR2 isoform FGFR2c loses ligand specificity, and able to bind with cognate FGF10 (Hajihosseini et al., 2009; Ibrahimi et al., 2001; Ibrahimi et al., 2004; Johnson and Wilkie, 2011; Yu et al., 2000). Interestingly, genetic knockdown of *Fgf10* in this FGFR2 mutant mouse can rescue the premature suture fusion (Hajihosseini et al., 2009). FGF signaling pathway members have not been linked to RAB23-mediated trafficking. However, a study suggests that RAB23 resides in the plasma membrane and it is proposed to be involved in endocytosis (Evans et al., 2003). It has been shown that RAB23 transports dopamine receptors to the primary cilium (Leaf and Von Zastrow, 2015) and in this context, RAB23 might be involved in ligand-receptor turnover.

It was first evident that RAB23 is a negative regulator of hedgehog (*Hh*) signaling in mouse neural tube development (Eggenschwiler et al., 2001) and further studies showed that RAB23 negatively regulates GLI1 in Su(Fu) dependant manner in which RAB23 expression reduces nuclear localization of *Gli1* transcription factor (Chi et al., 2012). *Gli1* positive cells have been identified in the suture as the main source of mesenchymal stem cells and *Gli1* plays critical role in maintaining suture patency (Zhao et al., 2015). The above mentioned findings collectively encouraged us to unveil the role of RAB23 in regulating *Hh* and *Gli1* in suture biology.

We highly agree with the reviewers that secondary mechanisms for these effects are possible. To understand such effects, we have performed new experiments and the findings are presented in Figure 6 (Figure 6E-H) where we show that inhibition of hedgehog transducer SMO by cyclopamine could not alter pERK1/2, rather inhibition of pERK1/2 could downregulate GLI1. Primary effects of RAB23 acting as a small GTP-ase during vesicle trafficking may regulate the turnover of FGFR and thereby FGF signaling, but there is no direct evidence for this. We have clarified this in the manuscript. The Figure 6E-H is incorporated in revised manuscript along with previously shown Figure 6A-D.

We agree that there is little robust evidence for the fundamental significance of *Fgf10* and *Pitx2*. These, however, good evidence that FGF signaling, in general, is involved in suture fusion and are good evidence for the involvement of *Hh* signaling via *Gli1*. Mutation in several *Hh* pathway members including *Gli1* result in premature suture fusion (Twigg et al., 2016; Zhao et al., 2015).

We thank reviewer for suggesting us to examine loss- or gain-of-function experiments the functionality of what could be the most critical disease causing gene or gene set. We have performed additional experiments to understand this, our findings are presented in new figures (Figure 6E-H, Figure 3H, Figure 1P-c and Figure 4A-B). All these figures now incorporated in revised manuscript. Here we showed that abrogation of GLI1 and pERK1/2 by combined action of cyclopamine and U0126, and cyclopamine alone indicates that RUNX2 is the most critical gene that lies downstream of both pERK1/2 and GLI1 (Figure 6E-H). Co-staining of RAB23 and RUNX2 (Figure 1P-c) showed overlapped expression in the osteogenic front where the EdU positive cells are more prominent. Interestingly, EdU positive cells also showed overlapped signals with RUNX2 in the osteogenic front (Figure 4A-B). In addition, we showed that stimulation of RAB23 deficient cells by FGF10, overexpressed RUNX2 expression (Figure 3H). Here, we shown two ways of RUNX2 regulation: (1) overactivating the FGFR signaling and by (2) abrogating the pERK1/2 signaling. Our lab previously shown that RUNX2 is a downstream candidate of *Hh* signaling and critical regulator of suture patency and therefore, loss of one allele of RUNX2 in overactivation *Hh* signaling mouse model of craniosynostosis rescued from premature suture fusion (Tanimoto et al., 2012). Current study thus showed well synergies and drawn the network of RAB23, FGFR, *Hh* and RUNX2 signaling in suture biology.

3) The authors checked premature suture fusion either by CT scanning or alizarin red-alzarin blue staining. Histological analysis, or at least coronal/sagittal sections of the suture CT images, are necessary to confirm the suture fusion. Furthermore, they need to co-stain markers, such as ALP and RUNX2, with their RNAscope analyses (e.g. Figure 1P, middle panel) and immunofluorescence staining images. It is currently difficult to differentiate the suture mesenchyme from the bone at these embryonic stages, which makes data interpretation difficult. Finally, to provide evidence for their conclusion that Rab23 lambdoid suture and MEF cells show increased osteoprogenitor cell proliferation, the authors must to co-localize Edu with osteoprogenitor cell markers in vivo, noting that Edu will label all proliferating cells.

We thank the reviewers for pointing out these important issues.

We have now provided a new figure, which confirms of coronal suture fusion in a micro-CT slice /section through the *Rab23*^-/-^ calvaria. We now incorporated this image into Figure 1 (Figure 1B-b, green arrow) in the revised manuscript.

In response to the co-stain marker issue, new immunohistochemical staining with RAB23 and RUNX2 has been carried out in Wt sections. Figure 1P-c shows RAB23 expression in the lambdoid suture (green) with RUNX2 co-stain marker (red). The figure shows RAB23 is expressed in the osteogenic front and RUNX2 is co-localized with RAB23 (yellow). The Figure 1P-c is incorporated in revised manuscript along with previously shown Rab23 RNAscope expression Figure 1P-b.

In response to whether the osteoprogenitor cells are proliferating we have now performed a RUNX2 immunohistochemistry on EdU-pulsed tissue sections (Wt and Rab23^-/-^). The new figure (Figure 4A-B) shows EdU-pulsed proliferating cells (red) in the Wt and Rab23 deficient sutures (E17.5) stained with osteoprogenitor marker RUNX2 (green). RUNX2 is co-localizes with EdU in the osteogenic front (yellow, white arrows) in both Wt and RAB23 deficient samples. The Figure 4A-B is incorporated in revised manuscript along with previously shown Figure 4C-J.

4) There is no supporting data for a downstream action of FGF10 although both FGF10 and PITX2 are increased in Rab23^-/-^ mice. The authors have only validated the in situ expression of Pitx2, not of Fgf10. There is published data showing that PITX2 is an upstream regulator of FGF10 during early organogenesis of other tissues. Data showing that FGF10 is downstream of PITX2 in suture development is therefore required to strengthen the conclusion.

We highly appreciate reviewers’ comments. Here a new figure (Figure 3H) added which indicating downstream action of FGF10. When RAB23 deficient cells were treated with or without FGF10 (500 ng/ml) for 2 and 4 hours followed by RT-qPCR analysis revealed FGF10 stimulation upregulates *Runx2* expression in both time points, indicating that *Runx2* is a downstream target of FGF10 signaling. The Figure 3H is incorporated in revised manuscript along with previously shown Figure 3.

In addition, we have now acknowledged in the manuscript text that we have not shown *Pitx2* regulation of FGF10 in the calvaria. Also, we downplayed this aspect by removing *Pitx2* from the schematic diagram (Figure 7). Hopefully, this improves the focus and clarifies the manuscript.

5) The authors noted an elevated expression of FGF10 and FGFR1b in Rab23^-/-^ mice, and concluded that increased FGFR1b was induced by FGF10. There is however the possibility that increased FGFR1b is a direct result of the Rab23 mutation. This possibility needs to be discussed or excluded experimentally.

This is a good point. We agree with the possibility that RAB23 could directly regulate FGFR1b. RAB23 is localized to the plasma membrane and are proposed to be involved in endocytosis and growth factor receptor recycling and turnover, and therefore, availability for ligand binding at the cell surface (Evans et al., 2003). This issue is now discussed in the Introduction and Discussion.

6) The authors found elevated pERK1/2, RUNX2, Hh and GLI1, as well as decreased p38 in Rab23^-/-^ mice. It however remains unclear whether the elevated pERK1/2 and pp38 are secondary to FGF signaling, and whether Hh and GLI1 upregulation is caused by phosphorylation of ERK1/2. Interestingly, published data shows that RAB23 can negatively regulate the Hh pathway directly. Therefore, it is critical to experimentally establish a direct role for RAB23 in changing gene expression. Promoter-reporter assays could be useful in this regard.

We appreciate this point. We agree that these aspects should be clarified. To test whether GLI1 upregulation is caused by phosphorylation of ERK1/2 alone, we used the SMO inhibitor cyclopamine to block canonical *Hh* signaling and also U0126 to block ERK1/2 (Individually and in combination with cyclopamine) in *Rab23*^-/-^ cells. New Figure 6A-H shows that U0126 and cyclopamine both individually can downregulate GLI1. This indicates that GLI1 is regulated through both canonical *Hh* signaling and also indirectly through pERK1/2. Cyclopamine did not downregulate pERK1/2, indicating that GLI1 resides downstream of pERK1/2.

The reviewers state that published data shows that RAB23 can negatively regulate the *Hh* pathway directly. Therefore, it is logical that the increased level of GLI1 in RAB23 mutant cells can in part be corrected by blocking canonical *Hh* signaling. Our investigation further unveiled that RUNX2 is downstream of both pERK1/2 and GLI1 (Figure 6E-H). These experiments combined with the data from Figures 2 and 3 establish a role for RAB23 in changing gene expression of FGF/MAPK signaling, *Hh*/GLI1 signaling and the osteoblast master regulator RUNX2. This data is also consistent with the findings that GTP bound block form of RAB23 reduces nuclear localization of GLI1 in hepatocellular carcinoma cell lines (Sun et al., 2012).

The Figure 6E-H is incorporated in revised manuscript along with previously shown Figure 6A-D.

7) The authors used pSmad1/5/8 to test whether the TGF-β superfamily was affected by the Rab23 mutation. Since pSmad1/5/8 is the readout of global BMP signaling, another readout of the TGF signaling pathway, pSmad2/3, should be tested.

This is an excellent point. We now provide new additional information in figure (Figure 3—figure supplement 1C-D) that pSMAD2/3 along with pSMAD1/5/8 are not differentially expressed in Wt and RAB23 deficient sutures. This figure is now updated in revised manuscript.

8) It is surprising that the epithelial, but not mesenchymal splice forms of FGFR1 are upregulated in suture and/or calvarial tissue. The authors should make some attempt to explain as to how or why this is the case if the sample should only contain mesenchymal cells.

We agree that it is surprising that the FGFR1b splice form, which is normally expressed in epithelia, is upregulated in the mesenchymal suture. However, this is not without precedent. While FGFRc isoforms are predominantly expresses in the mesenchyme and b isoforms in the epithelium, their expression is not mutually exclusive and FGFR1b is known to be expressed by late embryonic murine lung mesenchymal cells and also in the zebrafish in the sutural mesenchyme (Al Alam et al., 2015; Rice et al., 2003; Rice et al., 2004; Topczewska et al., 2016). Under pathological/force conditions mis-expression of splice forms can occur. Occasionally, Apert craniosynostosis syndrome is caused by heterozygous Alu-insertions or large deletions in the FGFR2c domalin. These rare mutations induce abnormal expression of the FGFR2b splice form alongside FGFR2c splice form in mesenchymal cells. This permits cells to respond to both b- and c-activating FGF ligands including FGF10 (Bochukova et al., 2009; Oldridge et al., 1999).

We have changed the manuscript text to elaborate on this surprising finding.

References:Fuller, K., O'Connell, J. T., Gordon, J., Mauti, O., and Eggenschwiler, J. (2014). Rab23 Helms, J. A., Cordero, D., and Tapadia, M. D. (2005). New insights into craniofacial morphogenesis. Development (Cambridge, England), 132(5), 851-861. doi:132/5/851 [pii]Leaf, A., and Von Zastrow, M. (2015). Dopamine receptors reveal an essential role of IFT-B, KIF17, and Rab23 in delivering specific receptors to primary cilia. *eLife, 4*, 10.7554/*eLife*.06996. doi:10.7554/*eLife*.06996 [doi]Liu, B., Lu, Y., Wang, Y., Ge, L., Zhai, N., and Han, J. (2019). A protocol for isolation and identification and comparative characterization of primary osteoblasts from mouse and rat calvaria. Cell and Tissue Banking, 20(2), 173-182. doi:10.1007/s10561-019-09751-0 [doi]Malaval, L., Liu, F., Roche, P., and Aubin, J. E. (1999). Kinetics of osteoprogenitor proliferation and osteoblast differentiation in vitro. Journal of Cellular Biochemistry, 74(4), 616-627. doi:10.1002/(SICI)1097-4644(19990915)74:43.0.CO;2-Q [pii]Sun, H. J., Liu, Y. J., Li, N., Sun, Z. Y., Zhao, H. W., Wang, C.,... Huang, S. H. (2012). Sublocalization of Rab23, a mediator of sonic hedgehog signaling pathway, in hepatocellular carcinoma cell lines. Molecular Medicine Reports, 6(6), 1276-1280. doi:10.3892/mmr.2012.1094 [doi]Tanimoto, Y., Veistinen, L., Alakurtti, K., Takatalo, M., and Rice, D. P. (2012). Prevention of premature fusion of calvarial suture in GLI-kruppel family member 3 (Gli3)-deficient mice by removing one allele of runt-related transcription factor 2 (Runx2). The Journal of Biological Chemistry, 287(25), 21429-21438. doi:10.1074/jbc.M112.362145 [doi]